# Recognition of discrete export signals in early flagellar subunits during bacterial type III secretion

Owain J Bryant[†], Paraminder Dhillon[‡], Colin Hughes, Gillian M Fraser*

Department of Pathology, University of Cambridge, Cambridge, United Kingdom

**Abstract** Type III Secretion Systems (T3SS) deliver subunits from the bacterial cytosol to nascent cell surface flagella. Early flagellar subunits that form the rod and hook substructures are unchaperoned and contain their own export signals. A gate recognition motif (GRM) docks them at the FlhBc component of the FlhAB-FliPQR export gate, but the gate must then be opened and subunits must be unfolded to pass through the flagellar channel. This induced us to seek further signals on the subunits. Here, we identify a second signal at the extreme N-terminus of flagellar rod and hook subunits and determine that key to the signal is its hydrophobicity. We show that the two export signal elements are recognised separately and sequentially, as the N-terminal signal is recognised by the flagellar export machinery only after subunits have docked at $FlhB_C$ via the GRM. The position of the N-terminal hydrophobic signal in the subunit sequence relative to the GRM appeared to be important, as a FlgD deletion variant ($FlgD_{short}$), in which the distance between the N-terminal signal and the GRM was shortened, 'stalled' at the export machinery and was not exported. The attenuation of motility caused by $FlgD_{short}$ was suppressed by mutations that destabilised the closed conformation of the FlhAB-FliPQR export gate, suggesting that the hydrophobic N-terminal signal might trigger opening of the flagellar export gate.

**\*For correspondence:**
gmf25@cam.ac.uk

**Present address:** [†]Department of Biochemistry, University of Oxford, Oxford, United Kingdom; [‡]The FEBS Journal Editorial Office, Cambridge, United Kingdom

## Editor's evaluation

Herein, the authors show that the flagellar Type III Secretion System recognize sequentially two discrete export signals, 1. initial docking and 2. subsequent opening of the export gate, for bacterial flagella biogenesis. This important and elegantly designed study elucidates a key step in solving the long-standing question of how export substrates are recognized by the type III secretion system.

## Introduction

Type III Secretion Systems (T3SS) are multi-component molecular machines that deliver protein cargo from the bacterial cytosol either to their site of assembly in cell surface flagella or virulence factor injectisomes, or directly to their site of action in eukaryotic target cells or the extracellular environment (*Evans et al., 2014a*; *Deng et al., 2017*; *Büttner and He, 2009*; *Konkel et al., 2004*; *Dongre et al., 2018*). The flagellar T3SS (fT3SS) directs the export of thousands of structural subunits required for the assembly and operation of flagella, rotary nanomotors for cell motility that extend from the bacterial cell surface (*Evans et al., 2014a*; *Evans et al., 2014b*). Newly synthesised subunits of the flagellar rod, hook and filament are targeted to the fT3SS, where they are unfolded and translocated across the cell membrane, powered by the protonmotive force and ATP hydrolysis, into an external export channel that spans the length of the nascent flagellum (*Minamino and Namba, 2008*; *Paul et al., 2008*). During flagellum biogenesis, when the rod/hook structure reaches its mature length, the fT3SS switches export specificity from recognition of 'early' rod/hook subunits to 'late' subunits for

filament assembly (***Williams et al., 1996***; ***Fraser et al., 2003***). This means that early and late flagellar subunits must be differentiated by the fT3SS machinery to ensure that they are exported at the correct stage of flagellum biogenesis. This is achieved, in part, by targeting subunits to the export machinery at the right time using a combination of export signals in the subunit mRNA and/or polypeptide. T3SS substrates contain N-terminal signals for targeting to the export machinery, however they do not share a common peptide sequence (***Kuwajima et al., 1989***; ***Minamino and Macnab, 1999***; ***Kornacker and Newton, 1994***; ***Evans et al., 2013***; ***Végh et al., 2006***). In addition, some substrates are piloted to the T3SS machinery by specific chaperones (***Wattiau et al., 1994***; ***Fraser et al., 1999***; ***Thomas et al., 2004***; ***Akeda and Galán, 2005***; ***Bange et al., 2010***).

The core export components of the fT3SS are evolutionarily related to those of the virulence injectisome, with which they share considerable structural and amino acid sequence similarity (***Kuhlen et al., 2018***; ***Abrusci et al., 2012***; ***Eichelberg et al., 1994***; ***Johnson et al., 2019***). The flagellar export machinery comprises an ATPase complex (FliHIJ) located in the cytoplasm, peripheral to the membrane. Immediately above the ATPase is a nonameric ring formed by the cytoplasmic domain of FlhA (FlhA$_C$), which functions as a subunit docking platform (***Bange et al., 2010***; ***Kinoshita et al., 2013***; ***Bryant et al., 2021***; ***Xing et al., 2018***). A recent cryo-ET map indicates that the FlhA family have a sea-horse-like structure, in which FlhAc forms the 'body' and the FlhA N-terminal region (FlhA$_N$) forms the 'head', which is fixed in the plane of the membrane (***Butan et al., 2019***). FlhA$_N$ wraps around the base of a complex formed by FliPQR and the N-terminal sub-domain of FlhB (FlhB$_N$), and together these form the FlhAB-FliPQR export gate that connects the cytoplasm to the central channel in the nascent flagellum, which is contiguous with the extracellular environment (***Kuhlen et al., 2018***; ***Abrusci et al., 2012***; ***Eichelberg et al., 1994***; ***Johnson et al., 2019***; ***Bryant et al., 2021***; ***Xing et al., 2018***; ***Butan et al., 2019***; ***Mizuno et al., 2011***). FlhB$_N$ is connected via a linker (FlhB$_{CN}$) to the cytoplasmic domain of FlhB (FlhB$_C$), which is thought to sit between the FlhA$_N$ and FlhA$_C$ rings, where it functions as a docking site for early flagellar subunits (***Evans et al., 2013***; ***Kuhlen et al., 2018***; ***Butan et al., 2019***).

The 'early' flagellar subunits that assemble to form the rod and hook substructures are not chaperoned: instead, the signals for targeting and export are found within the early subunits themselves. We have shown that of one of these signals is a small hydrophobic sequence termed the gate recognition motif (GRM), which is essential for early subunit export (***Evans et al., 2013***). This motif binds a surface exposed hydrophobic pocket on FlhBc (***Evans et al., 2013***). Once subunits reach the export machinery, they must be unfolded before they can pass through the narrow channel formed by FliPQR-FlhB$_N$ into the central channel of the nascent flagellum, through which the subunits transit until they reach the tip and fold into the structure (***Evans et al., 2014b***; ***Kuhlen et al., 2018***). Structural studies suggest that FliPQR-FlhB$_N$ adopts an energetically favourable closed conformation, possibly to maintain the membrane permeability barrier (***Kuhlen et al., 2018***; ***Johnson et al., 2019***; ***Ward et al., 2018***; ***Kuhlen et al., 2020***). This suggests that there must be a mechanism to trigger opening of the export gate when subunits dock at cytosolic face of the flagellar export machinery.

Here, we sought to identify new export signals within flagellar rod/hook subunits, using the hook-cap subunit FlgD as a model export substrate. We show that the extreme N-terminus of rod/hook subunits contains a hydrophobic export signal and investigate its functional relationship to the subunit gate recognition motif (GRM).

## Results

### Identification of a hydrophobic export signal at the N-terminus of FlgD

The N-terminal region of flagellar rod and hook subunits is required for their export (***Minamino and Macnab, 1999***; ***Evans et al., 2013***). Using the flagellar hook-cap protein FlgD as a model rod/hook subunit, we sought to identify specific export signals within the N-terminus. A screen of ten FlgD variants containing internal five-residue scanning deletions in the first 50 residues (although FlgDΔ2–5 is a four-residue deletion, retaining the initial methionine) identified just two variants defective for export into culture supernatant (***Figure 1A***). Loss of residues 2–5 caused a significant reduction in export, as did deletion of residues 36–40, although to a lesser extent (***Figure 1A***; ***Evans et al., 2013***). We have shown that FlgD residues 36–40 are the gate recognition motif (GRM) required for transient subunit

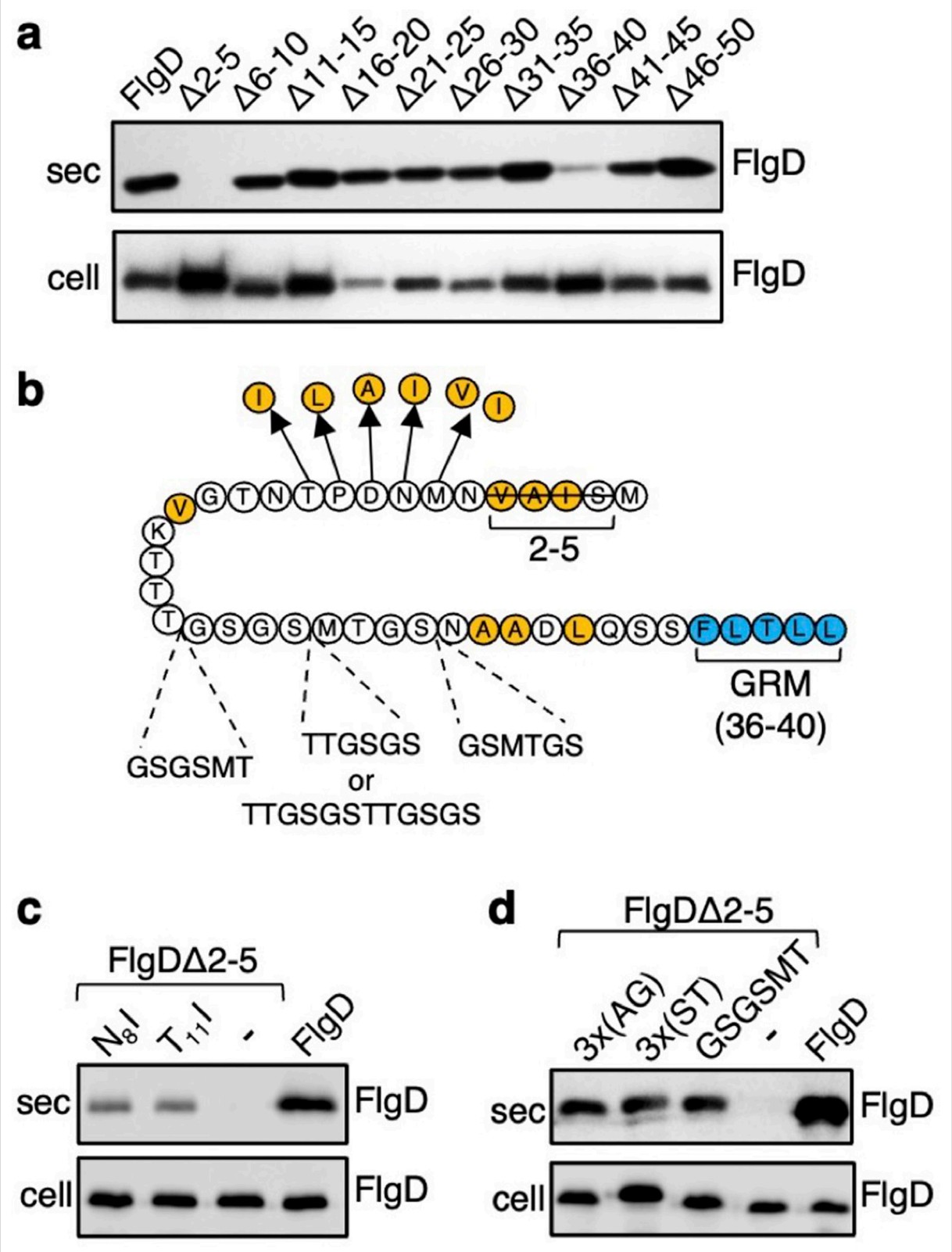

**Figure 1.** Screening for export-defective FlgD variants.

(**a**) Whole cell (cell) and supernatant (sec) proteins from late exponential phase cultures of a *Salmonella flgD* null strain expressing plasmid-encoded wild type FlgD (FlgD) or its variants (Δ2–5, Δ6–10, Δ11–15, Δ16–20, Δ21–25, Δ26–30, Δ31–35, Δ36–40, Δ41–45 or Δ46–50) were separated by SDS (15%)-PAGE and analysed by immunoblotting with anti-FlgD polyclonal antisera. (**b**) A schematic displaying all intragenic suppressor mutations within amino

*Figure 1 continued on next page*

*Figure 1 continued*

acids 1–40 of FlgD isolated from the FlgDΔ2–5 variant. Small non-polar residues are highlighted in orange. All suppressor mutations were located between the gate-recognition motif (GRM, blue) and the extreme N-terminus, and can be separated into two classes: insertions or duplications that introduced additional sequence between valine-15 and the gate-recognition motif, or missense mutations that re-introduce small non-polar residues at the N-terminus. All intragenic suppressors isolated from FlgDΔ2–5 are displayed in this figure. (**c**) Whole cell (cell) and supernatant (sec) proteins from late exponential-phase cultures of *Salmonella flgD* null strains expressing plasmid-encoded: suppressor mutants isolated from the FlgDΔ2–5 variant (FlgDΔ2–5-N$_8$I or FlgDΔ2–5-T$_{11}$I), FlgDΔ2–5 variant (-) or wild type FlgD (FlgD) were separated by SDS (15%)-PAGE and analysed by immunoblotting with anti-FlgD polyclonal antisera. (**d**) Whole cell (cell) and supernatant (sec) proteins from late exponential phase cultures of *Salmonella flgD* null strains expressing plasmid-encoded wild type FlgD (labelled FlgD), FlgDΔ2–5 (labelled as DΔ2–5) or variants of FlgDΔ2–5 containing between residues 19 and 20 a six-residue insertion of either small non-polar (AGAGAG) residues (labelled as 3 x(AG)), polar (STSTST) residues (labelled as 3 x(ST)), or the sequence from an isolated insertion suppressor mutant (GSGSMT) (labelled as GSGSMT), were separated by SDS (15%)-PAGE and analysed by immunoblotting with anti-FlgD polyclonal antisera.

The online version of this article includes the following source data and figure supplement(s) for figure 1:

**Source data 1.** Full-length protein western blot of secreted proteins relating to *Figure 1A*.

**Source data 2.** Full-length western blot of cellular proteins relating to *Figure 1A*.

**Source data 3.** Full-length western blot of cellular and secreted proteins relating to *Figure 1C* (bottom).

**Source data 4.** Full-length western blot of cellular proteins relating to *Figure 1D* (low exposure, bottom).

**Source data 5.** Full-length western blot of cellular proteins relating to *Figure 1D* (high exposure, bottom).

**Figure supplement 1.** Relative amounts of FlgD secreted into culture supernatants from a recombinant *Salmonella flgD* null strain expressing plasmid-based FlgDΔ2–5 suppressor mutants and wild type FlgD.

**Figure supplement 2.** Hydrophobicity of N-terminal sequences of flagellar rod and hook subunits.

**Figure supplement 3.** Deletion of the FliK N-terminal signal attenuates export.

**Figure supplement 3—source data 1.** Full-length protein western blot of cellular and secreted proteins relating to *Figure 1—figure supplement 1* (anti-myc).

**Figure supplement 3—source data 2.** Full-length protein western blot of cellular and secreted proteins relating to *Figure 1—figure supplement 1* (anti-FlhA).

**Figure supplement 3—source data 3.** Full-length protein western blot of cellular and secreted proteins relating to *Figure 1—figure supplement 1* (anti-FlgN).

**Figure supplement 4.** Swimming motility of FlgDdelta2-5 suppressor mutant strains.

**Figure supplement 5.** Control blots for *Figure 1C-D* showing that membrane embedded FlhA and cytoplasmic FlgN proteins are expressed (cell) but absent from supernatant fractions (sec).

**Figure supplement 5—source data 1.** Full-length protein western blot of cellular and secreted proteins relating to *Figure 1—figure supplement 5A* (anti-FlhA (top) and anti-FlgN (bottom)).

**Figure supplement 5—source data 2.** Full-length protein western blot of cellular and secreted proteins relating to *Figure 1—figure supplement 5B* (anti-FlhA (top) and anti-FlgN (bottom)).

docking at the FlhB$_C$ component of the export gate (*Evans et al., 2013*). The results suggest that the extreme N-terminus might also be important for interaction with the export machinery.

To gain insight into the putative new signal, we screened for intragenic suppressor mutations that could restore export of the FlgDΔ2–5 variant. A *Salmonella flgD* null strain expressing *flgD*Δ2–5 in trans was inoculated into soft-tryptone agar and incubated until 'spurs' of motile cell populations appeared. Sequencing of *flgD*Δ2–5 alleles from these motile populations identified ten different intragenic gain-of-function mutations. These could be separated into two classes (*Figure 1B*, *Figure 1—figure supplement 1*).

The first class of motile revertants carried *flgD*Δ2–5 alleles with missense mutations that introduced small non-polar residues at the extreme N-terminus of FlgDΔ2–5 (*Figure 1B*). Deletion of residues 2–5 (²SIAV⁵) had removed all small non-polar amino acids from the first ten residue region of FlgD, effectively creating a new N-terminus containing a combination of polar, charged or large non-polar residues (*Figure 1—figure supplement 2*). Analysis of other flagellar rod and hook subunit primary sequences revealed that in every case their native N-terminal regions contain small non-polar residues positioned upstream of the gate recognition motif (GRM residues 36–40; *Figure 1—figure supplement 2*), indicating that hydrophobicity may be key to the function of the N-terminal export signal. Removal of non-polar residues from the extreme N-terminus of the secreted hook-length control

protein, FliK, attenuated its export, indicating that the hydrophobic N-terminal signal is required for export of early subunits (*Figure 1—figure supplement 3*).

Export assays performed with two representative motile revertant strains carrying *flgDΔ2–5* variants with gain-of-function point mutations, those encoding FlgDΔ2–5-$N_8$I and FlgDΔ2–5-$T_{11}$I, revealed that export of these subunits had recovered to ~50% of the level observed for wild type FlgD (*Figure 1C*, *Figure 1—figure supplements 4–5*).

The second class of motile revertants carried *flgDΔ2–5* alleles that had acquired duplications or insertions introducing at least six additional residues between the FlgDΔ2–5 N-terminus and the gate recognition motif (GRM; *Figure 1B*). It seemed possible that these insertions/duplications might have restored subunit export either by insertion of amino acids that could function as a 'new' hydrophobic export signal, or by restoring the position of an existing small hydrophobic residue or sequence relative to the GRM.

To assess these possibilities, we tested whether export of FlgDΔ2–5 could be recovered by inserting either polar ($^{19}$STSTST$^{20}$) or small non-polar ($^{19}$AGAGAG$^{20}$) residues in the FlgDΔ2–5 N-terminal region at a position equivalent to one of the suppressing duplications ($^{19}$GSGSMT$^{20}$; *Figure 1B and D*, *Figure 1—figure supplements 4–5*). We reasoned that if suppression by the additional sequence had been caused by repositioning an existing small hydrophobic amino acid relative to the GRM, then any insertional sequence (polar or non-polar) would restore export, while if suppression had resulted from insertion of a 'new' export signal, then either the polar STSTST or non-polar AGAGAG, but not both, could be expected to restore export.

We found that both the engineered FlgD variants (FlgDΔ2-5-$^{19}$AGAGAG$^{20}$ and FlgDΔ2-5-$^{19}$STSTST$^{20}$) were exported from a *Salmonella flgD* null strain as effectively as the gain-of-function mutant FlgDΔ2-5-$^{19}$GSGSMT$^{20}$ isolated from the suppressor screen (*Figure 1D*). This suggests that the insertions had repositioned a sequence in the FlgDΔ2–5 N-terminus relative to the GRM to overcome the loss of small hydrophobic residues.

## The position of the hydrophobic export signal relative to the gate recognition motif is critical for Rod and Hook subunit export

The intragenic suppressor experiments indicated that FlgD export requires a hydrophobic export signal towards the N-terminus and that the position of this hydrophobic signal relative to the previously described GRM is important. Sequence analysis of the gain-of-function FlgDΔ2–5 insertion variants revealed that the insertions were all located between the GRM and valine$_{15}$ ($V_{15}$; *Figure 1B*). We reasoned that they repositioned valine$_{15}$ relative to the GRM, such that it could perform the function of the N-terminal hydrophobic signal lost in FlgDΔ2–5. To test this view, we replaced $V_{15}$ by alanine in the gain-of-function variant FlgDΔ2-5-$^{19}$(GSGSMT)$^{20}$ and assayed its export in the *Salmonella flgD* null (*Figure 2A*, *Figure 2—figure supplement 1*). Unlike the *flgD* null strain producing either the parental FlgDΔ2-5-$^{19}$(GSGSMT)$^{20}$ or wild type FlgD, the *flgD* null carrying variant FlgDΔ2-5-$^{19}$(GSGSMT)$^{20}$-$V_{15}$A was non-motile, reflecting the variant's failure to export (*Figure 2A*, *Figure 2—figure supplement 1*). This suggests that the $V_{15}$ residue had indeed compensated for the missing N-terminal hydrophobic signal.

By screening for intragenic suppressors of the motility defect associated with FlgDΔ2-5-$^{19}$(GSGSMT)$^{20}$-$V_{15}$A, four gain-of-function missense mutations were identified, $M_7$I, $D_9$A, $T_{11}$I and $G_{14}$V. All these had introduced small hydrophobic residues, all positioned at least 27 residues upstream of the GRM. These FlgDΔ2-5-$^{19}$(GSGSMT)$^{20}$-$V_{15}$A gain-of-function variants restored motility to the *Salmonella flgD* null strain and were exported at levels similar to wildtype FlgD and FlgDΔ2-5-$^{19}$(GSGSMT)$^{20}$ (*Figure 2A*, *Figure 2—figure supplement 1*). These data confirm the importance of small non-polar residues positioned upstream of the GRM.

Our results so far had indicated that the position of the FlgD N-terminal hydrophobic export signal relative to the GRM was critical and suggested that, for export to occur efficiently, at least 26 residues must separate the hydrophobic signal and the GRM (*Figure 1C*, *Figure 1—figure supplement 4*). In the primary sequences of all *Salmonella* flagellar rod/hook subunits the GRM is positioned ≥30 amino acids downstream of the subunit N-terminus (*Figure 1—figure supplement 4*), suggesting that separation of the two signals by a minimum number of residues might be a common feature among early flagellar subunits. To test this, a suite of engineered *flgD* alleles was constructed that encoded FlgD variants in which wildtype residues 9–32 were replaced with between one and four repeats

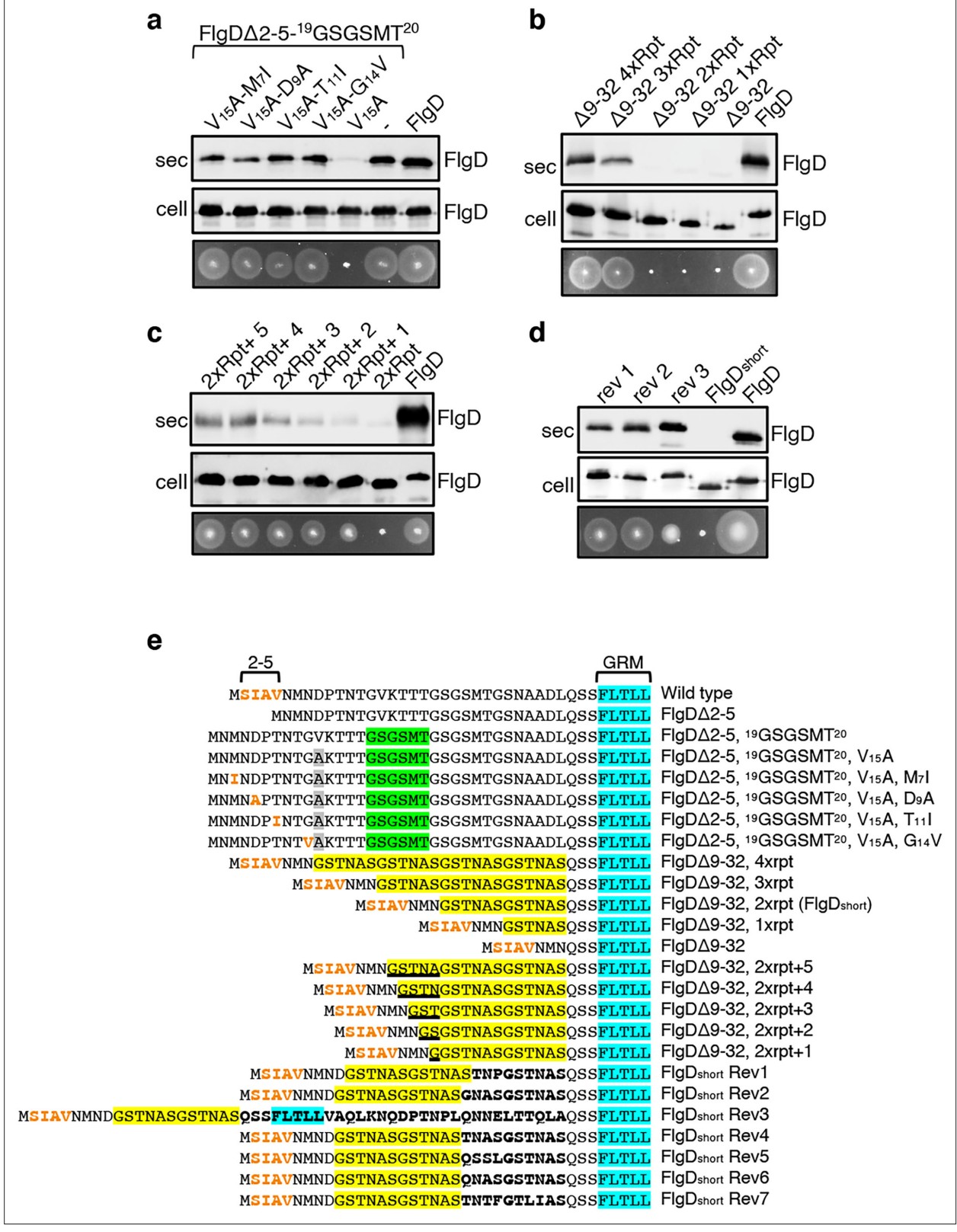

**Figure 2.** Export of FlgD variants in which the position of the hydrophobic export signal is varied relative to the gate recognition motif (GRM). Whole cell (cell) and supernatant (sec) proteins from late exponential-phase cultures of a *Salmonella flgD* null strain expressing plasmid encoded suppressor mutants isolated from the FlgDΔ2-5-19(GSGSMT)20 $V_{15}A$ variant ($V_{15}A$-$M_7$I, $V_{15}A$-$D_9A$, $V_{15}A$-$T_{11}$I, $V_{15}A$-$G_{14}$V), their parent FlgD variant FlgDΔ2-5-19(GSGSMT)20 $V_{15}A$ (labelled as $V_{15}A$), FlgDΔ2-5-19(GSGSMT)20 (labelled as -) or wild type FlgD (FlgD) were separated by SDS (15%)-PAGE and

*Figure 2 continued on next page*

*Figure 2 continued*

analysed by immunoblotting with anti-FlgD polyclonal antisera. Swimming motility (bottom panel; 0.25% soft tryptone agar) of the same strains were carried out at 37 °C for 4–6 hours. (**b**). Whole cell (cell) and supernatant (sec) proteins from late exponential-phase cultures of a *Salmonella flgD* null strain expressing plasmid-encoded wild type FlgD (labelled as FlgD), FlgDΔ9–32 or its variants in which residues 9–32 were replaced by between one and four six-residue repeats of Gly-Ser-Thr-Asn-Ala-Ser (GSTNAS): (Δ9–32 4xRpt, Δ9–32 3xRpt, Δ9–32 2xRpt or Δ9–32 1xRpt) were separated by SDS (15%)-PAGE and analysed by immunoblotting with anti-FlgD polyclonal antisera. Swimming motility (bottom panel; 0.25% soft tryptone agar) of the same strains were carried out at 37 °C for 4–6 hours. (**c**). Whole cell (cell) and supernatant (sec) proteins from late exponential-phase cultures of a *Salmonella flgD* null strain expressing plasmid-encoded wild type FlgD (labelled as FlgD), a FlgD variant in which residues 9–32 were replaced by two repeats of a six-residue sequence Gly-Ser-Thr-Asn-Ala-Ser (labelled as 2xRpt) or its variants containing between one and five additional residues inserted directly after the two repeats (labelled as 2xRpt + 1, 2xRpt + 2, 2xRpt + 3, 2xRpt + 4 or 2xRpt + 5) were separated by SDS (15%)-PAGE and analysed by immunoblotting with anti-FlgD polyclonal antisera. Swimming motility (bottom panel; 0.25% soft tryptone agar) of the same strains were carried out at 37 °C for 4–6 hr. (**d**). Whole cell (cell) and supernatant (sec) proteins from late exponential-phase cultures of a *Salmonella flgD* null strain expressing plasmid-encoded wild type FlgD (labelled as FlgD), a FlgD variant in which residues 9–32 were replaced by two repeats of a six-residue sequence Gly-Ser-Thr-Asn-Ala-Ser (labelled as FlgD$_{short}$) or suppressor mutants isolated from this strain (labelled as rev1, rev2 or rev3) were separated by SDS (15%)-PAGE and analysed by immunoblotting with anti-FlgD polyclonal antisera. Swimming motility (bottom panel; 0.25% soft tryptone agar) of the same strains were carried out at 37 °C for 4–6 hr. (**e**). N-terminal sequences of wild type FlgD and its variants aligned to their gate-recognition motif (GRM; blue). The following sequence features or residues are displayed: The N-terminal hydrophobic signal (residue 2–5; orange), the Gly-Ser-Gly-Ser-Met-Thr (GSGSMT) insertion (green) isolated from the FlgDΔ2–5 suppressor screen, the valine-15 to alanine mutation (grey), small non-polar mutations (M7I, D9A, T11I, G14V; orange) isolated from the FlgDΔ2–5, $^{19}$GSGSMT$^{20}$ suppressor screen, FlgDΔ9–32 and its variants in which residues 9–32 are replaced with one, two, three or four repeats of a six-residue sequence Gly-Ser-Thr-Asn-Ala-Ser (GSTNAS; yellow), FlgDΔ9–32, 2xrpt (hereafter termed FlgD$_{short}$) containing five, four, three, two, or one additional residues (underlined) inserted between the GRM and N-terminal hydrophobic signal, and suppressor mutants (Rev 1–7) isolated from FlgD$_{short}$ that introduced additional residues (bold) between the N-terminal hydrophobic signal (orange) and the gate-recognition motif (blue).

The online version of this article includes the following source data and figure supplement(s) for figure 2:

**Source data 1.** Full-length protein western blot of cellular and secreted proteins relating to *Figure 2A*.

**Source data 2.** Full-length protein western blot of cellular and secreted proteins relating to *Figure 2B* (bottom).

**Source data 3.** Full-length protein western blot of cellular proteins relating to *Figure 2C* (bottom).

**Source data 4.** Full-length protein western blot of secreted proteins relating to *Figure 2C* (bottom).

**Source data 5.** Full-length protein western blot of cellular proteins relating to *Figure 2D* (bottom).

**Source data 6.** Full-length protein western blot of secreted proteins relating to *Figure 2D* (bottom, left).

**Figure supplement 1** Control blots for Figure 2A-C showing that membrane-embedded FlhA and cytoplasmic FlgN are expressed (cell) but absent from supernatant fractions (sec).

**Figure supplement 1—source data 1.** Full-length protein western blot of cellular and secreted proteins relating to *Figure 2—figure supplement 1A* (anti-FlhA).

**Figure supplement 1—source data 2.** Full-length protein western blot of cellular proteins relating to *Figure 2—figure supplement 1A* (anti-FlgN).

**Figure supplement 1—source data 3.** Full-length protein western blot of secreted proteins relating to *Figure 2—figure supplement 1B* (anti-FlgN).

**Figure supplement 1—source data 4.** Full-length protein western blot of cellular and secreted proteins relating to *Figure 2—figure supplement 1B* (anti-FlhA, top).

**Figure supplement 1—source data 5.** Full-length protein western blot of cellular and secreted proteins relating to *Figure 2—figure supplement 1B* (anti-FlgN, bottom).

**Figure supplement 1—source data 6.** Full-length protein western blot of cellular and secreted proteins relating to *Figure 2—figure supplement 1C* (anti-FlhA (top) and anti-FlgN (bottom)).

**Figure supplement 1—source data 7.** Full-length protein western blot of cellular and secreted proteins relating to *Figure 2—figure supplement 1D* (anti-FlhA (top) and anti-FlgN (bottom)).

**Figure supplement 2.** Control blots for *Figure 3B-C* showing that membrane embedded FlhA and cytoplasmic FlgN proteins (cell) are expressed but absent from supernatant fractions (sec).

**Figure supplement 2—source data 1.** Full-length protein western blot of cellular and secreted proteins relating to *Figure 2—figure supplement 2A* (anti-FlhA, bottom).

**Figure supplement 2—source data 2.** Full length protein western blot of cellular and secreted proteins relating to *Figure 2—figure supplement 2A* (anti-FlgN, bottom).

**Figure supplement 2—source data 3.** Full-length protein western blot of cellular and secreted proteins relating to *Figure 2—figure supplement 2B* (anti-FlhA, bottom).

**Figure supplement 2—source data 4.** Full-length protein western blot of cellular and secreted proteins relating to *Figure 2—figure supplement 2B* (anti-FlgN, bottom).

of the six amino acid sequence Gly-Ser-Thr-Asn-Ala-Ser (GSTNAS). Swimming motility and export assays revealed that the minimum number of inserted GSTNAS repeats that could support efficient FlgD export was three, equivalent to separation of the hydrophobic N-terminal signal and the GRM by 24 residues (*Figure 2B*). Below this threshold, FlgD export and swimming motility were strongly attenuated (*Figure 2B*). A further set of recombinant *flgD* alleles was constructed, which encoded FlgDΔ9–32 variants carrying two GSTNAS repeats (hereafter termed FlgD$_{short}$) directly followed by between one and five additional residues (*Figure 2E*). Motility and FlgD export increased incrementally with the addition of each amino acid (*Figure 2C*, *Figure 2—figure supplement 1*). The data indicate that a low level of FlgD export is supported when the hydrophobic N-terminal signal ($_{2}$SIAV$_{5}$) and the GRM ($_{36}$FLTLL$_{40}$) are separated by 19 residues, with export efficiency and swimming motility increasing as separation of the export signals approaches an optimal 24 residues.

To further establish the requirement for a minimum number of residues between the hydrophobic N-terminal signal and the GRM, we screened for intragenic suppressor mutations that could restore swimming motility in a *flgD* null strain producing FlgD$_{short}$. Sequencing of *flgD$_{short}$* alleles from motile revertant strains identified seven gain-of-function mutations that introduced additional residues between the hydrophobic N-terminal signal and the GRM (*Figure 2—figure supplement 1*).

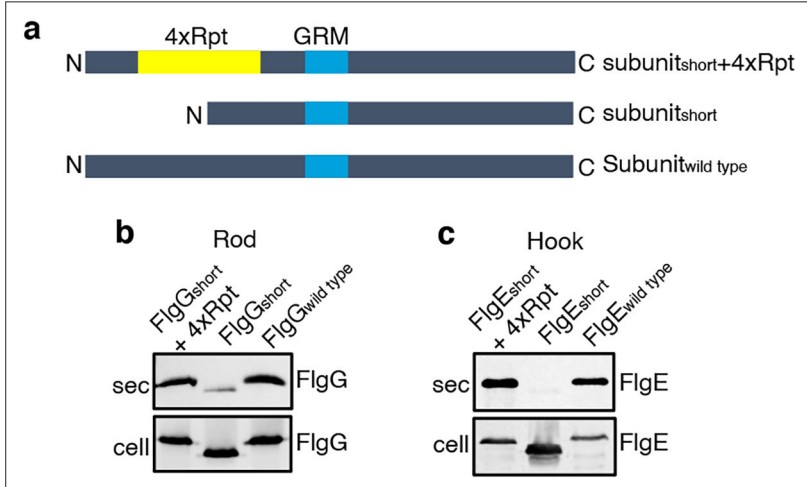

**Figure 3.** Effect of the relative position of the N-terminus and GRM on the export of other rod and hook subunits. (**a**) Schematic representation of a wild-type subunit (labelled as subunit$_{wild\ type}$), a subunit containing a deletion of sequence from between the N-terminus and GRM (labelled as subunit$_{short}$) and a subunit in which the deleted sequence was replaced by four repeats of a six-residue sequence Gly-Ser-Thr-Asn-Ala-Ser (yellow, labelled as subunit$_{short}$ +4 Rpt). (**b**). Whole cell (cell) and supernatant (sec) proteins from late exponential-phase cultures of a *Salmonella flgE* null strain expressing plasmid-encoded wild type FlgG (labelled as FlgG$_{wild\ type}$), a FlgG variant in which residues 11–35 were deleted (labelled as FlgG$_{short}$) or a FlgG variant in which residues 11–35 were replaced by four repeats of a six-residue sequence Gly-Ser-Thr-Asn-Ala-Ser (labelled as FlgG$_{short}$ + 4 Rpt). All FlgG variants were engineered to contain an internal 3xFLAG tag for immunodetection. Proteins were separated by SDS (15%)-PAGE and analysed by immunoblotting with anti-FLAG monoclonal antisera. (**c**). Whole cell (cell) and supernatant (sec) proteins from late exponential-phase cultures of a *Salmonella flgD* null strain expressing plasmid-encoded wild type FlgE (labelled as FlgE$_{wild\ type}$), a FlgE variant in which residues 9–32 were deleted (labelled as FlgE$_{short}$) or a FlgE variant in which residues 9–32 were replaced by four repeats of a six-residue sequence Gly-Ser-Thr-Asn-Ala-Ser (labelled as FlgE$_{short}$ + 4 Rpt). All FlgE variants were engineered to contain an internal 3xFLAG tag for immunodetection. Proteins were separated by SDS (15%)-PAGE and analysed by immunoblotting with anti-FLAG monoclonal antisera.

The online version of this article includes the following source data for figure 3:

**Source data 1.** Full-length protein western blot of cellular and secreted proteins relating to *Figure 3B* (low exposure).

**Source data 2.** Full-length protein western blot of cellular and secreted proteins relating to *Figure 3B* (high exposure).

**Source data 3.** Full-length protein western blot of secreted proteins relating to *Figure 3C* (low exposure, top).

**Source data 4.** Full-length protein western blot of cellular proteins relating to *Figure 3C* (high exposure, bottom).

Swimming motility and FlgD export was assessed for three *flgD* null strains expressing representative *flgD*$_{short}$ gain-of-function variants and all showed increased FlgD subunit export and swimming motility compared to the *flgD* null expressing *flgD*$_{short}$ (*Figure 2D*, *Figure 2—figure supplement 1*). The data confirm that the position of the hydrophobic N-terminal signal relative to the GRM is critical for efficient FlgD subunit export.

To establish that this is a general requirement for the export of other rod and hook subunits, engineered alleles of *flgE* (hook) and *flgG* (rod) were constructed that encoded variants in which FlgE residues 9–32 or FlgG residues 11–35 were either deleted (FlgE$_{short}$ and FlgG$_{short}$) or replaced with four repeats of the sequence GSTNAS (*Figure 3*). As had been observed for FlgD$_{short}$, export of the FlgE$_{short}$ and FlgG$_{short}$ variants was severely attenuated compared to wild-type FlgE and FlgG (*Figure 3*, *Figure 2—figure supplement 2*). Furthermore, insertion of four GSTNAS repeats into FlgE$_{short}$ and FlgG$_{short}$ recovered subunit export to wild-type levels, indicating that the minimum separation of the hydrophobic N-terminal signal and the GRM is a feature throughout rod and hook subunits (*Figure 3*).

## Sequential engagement of the subunit GRM and hydrophobic N-terminal export signal by the flagellar export machinery

Having identified a new hydrophobic N-terminal export signal and established that its position relative to the GRM was critical, we next wanted to determine the order in which the signals were recognised/engaged by the export machinery. The signals might be recognised simultaneously, with both being

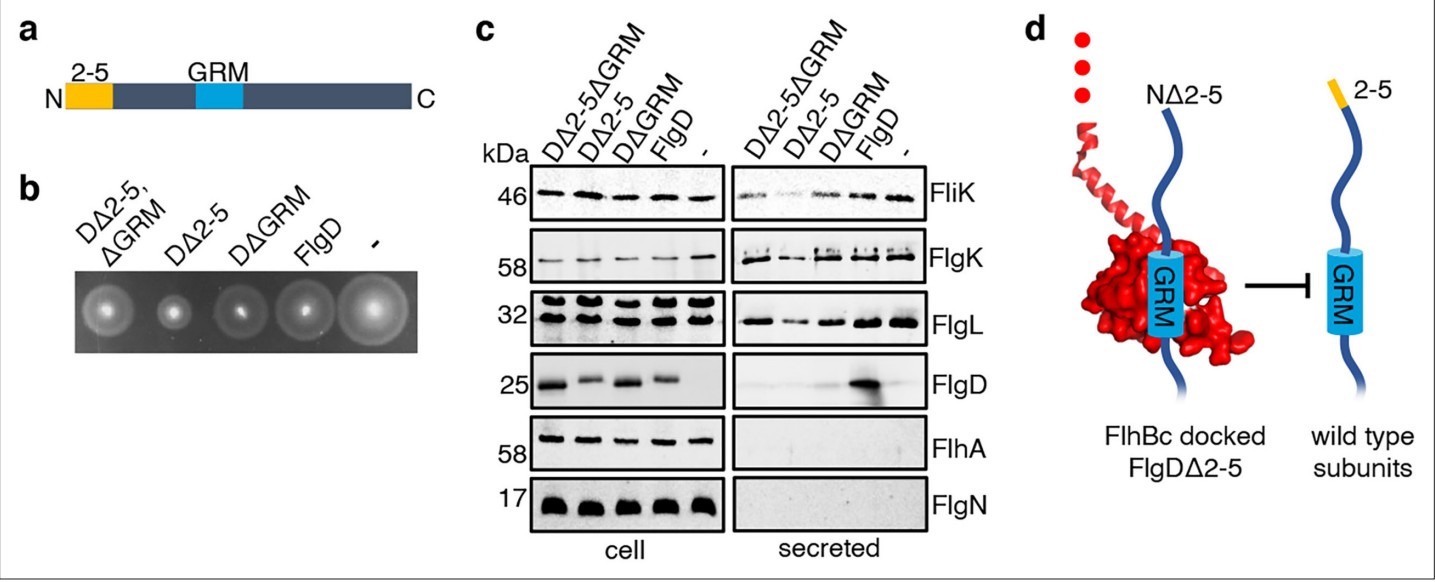

**Figure 4.** Effect on subunit export of overexpressed FlgDΔ2–5 and variants.
(a) Schematic representation of a FlgD subunit containing a N-terminal hydrophobic signal (orange, labelled as 2–5) and a gate-recognition motif (blue, labelled as GRM). (b) Swimming motility of a *Salmonella* Δ*recA* strain expressing plasmid-encoded wild type FlgD (FlgD), its variants (DΔ2-5ΔGRM, DΔ2–5 or DΔGRM) or empty pTrc99a vector (-). Motility was assessed in 0.25% soft-tryptone agar containing 100 µg/ml ampicillin and 100 µM IPTG and incubated for 4–6 hr at 37 °C. (c) Whole cell (cell) and secreted proteins (secreted) from late-exponential-phase cultures were separated by SDS (15%)-PAGE and analysed by immunoblotting with anti-FliK (hook ruler subunit), anti-FlgK and anti-FlgL (hook-filament junction subunits), anti-FlgD hook cap subunit, anti-FlhA (component of the export machinery) and anti-FlgN (export chaperone for FlgK and FlgL) polyclonal antisera. Apparent molecular weights are in kilodaltons (kDa).(d). A model depicting a FlgDΔ2–5 subunit (left) docked *via* its gate-recognition motif (GRM, blue) at the subunit binding pocket on FlhB$_C$ (PDB: 3B0Z *Kuhlen et al., 2020*, red), preventing wild type subunits (right) from docking at FlhB$_C$.

The online version of this article includes the following source data for figure 4:

**Source data 1.** Full-length protein western blot of cellular and secreted proteins relating to *Figure 4C* (anti-FliK, bottom).

**Source data 2.** Full-length protein western blot of cellular and secreted proteins relating to *Figure 4C* (anti-FlgK, top).

**Source data 3.** Full-length protein western blot of cellular and secreted proteins relating to *Figure 4C* (anti-FlgL).

**Source data 4.** Full-length protein western blot of cellular proteins relating to *Figure 4C* (anti-FlgD, top right).

**Source data 5.** Full-length protein western blot of secreted proteins relating to *Figure 4C* (anti-FlgD, bottom).

**Source data 6.** Full-length protein western blot of cellular and secreted proteins relating to *Figure 4C* (anti-FlhA (top) and anti-FlgN (bottom)).

required for initial entry of rod/hook subunits into the export pathway. Alternatively, they might be recognised sequentially. If this were the case, then a subunit variant that possessed the 'first' signal but was deleted for the 'second' signal might enter the export pathway but fail to progress, becoming stalled at a specific step to block the pathway and prevent export of wild type subunits. To test if FlgDΔ2–5 or FlgDΔGRM stalled in the export pathway, recombinant expression vectors encoding these variants or wild type FlgD were introduced into a *Salmonella* Δ*recA* strain that is wild type for flagellar export (*Figure 4*). We could then assess whether the variant FlgD constructs could interfere in trans with the wildtype flagellar export. To do this, we assessed the export of an early flagellar substrate, FliK, which controls the length of the flagellar hook and of the late export substrate, FlgK and FlgL, which together form a junction connecting the flagellar filament to the hook. We saw that FlgDΔ2–5 inhibited motility and export of the FliK, FlgK and FlgL flagellar subunits, whereas FlgDΔGRM did not (*Figure 4B and C*). The data indicate that FlgDΔ2–5 enters the flagellar export pathway and stalls at a critical point, blocking export. In contrast, FlgDΔGRM does not stall or block export.

To determine whether FlgDΔ2–5 stalls at a point before or after subunit docking at the FlhB$_C$ component of the flagellar export gate via the GRM, a recombinant vector encoding a FlgD variant in which both export signals were deleted (FlgDΔ2–5ΔGRM) was constructed. If loss of the hydrophobic N-terminal signal had caused subunits to stall after docking at FlhB$_C$, then additional deletion of the subunit GRM would relieve this block. Motility and subunit export assays revealed that the *Salmonella* Δ*recA* strain producing FlgDΔ2–5ΔGRM displayed swimming motility and levels of FliK and FlgK subunit export similar to cells producing FlgDΔGRM (*Figure 4B and C*). The data suggest that FlgDΔ2–5 stalls after docking at the FlhB$_C$ export gate, preventing docking of other early subunits.

It seemed possible that subunit docking via the GRM to the FlhB$_C$ export gate might position the hydrophobic N-terminal signal in close proximity to its recognition site on the export machinery. If this were the case, 'short' subunit variants containing deletions that decreased the number of residues between the hydrophobic N-terminal signal and the GRM might also stall at FlhB$_C$, and this stalling might be relieved by additional deletion of the GRM. To test this, recombinant expression vectors encoding 'short' subunit variants (FlgE$_{short}$ or FlgD$_{short}$), 'short' subunit variants additionally deleted for the GRM (FlgE$_{short}$ΔGRM or FlgD$_{short}$ΔGRM) or wild type FlgE or FlgD were introduced into a *Salmonella* Δ*recA* strain (*Figure 5*). Compared to the wild-type subunits expressed in trans, the 'short' subunits inhibited swimming motility and the export of other flagellar subunits (FliD, FliK, FlgK), whereas FlgE$_{short}$ΔGRM and FlgD$_{short}$ΔGRM did not (*Figure 5*). Taken together with the data presented in *Figure 4*, the results indicate that the subunit GRM and the hydrophobic N-terminal signal are recognised sequentially, with subunits first docking at FlhB$_C$ via the GRM, which positions the hydrophobic N-terminal signal for subsequent interactions with the export machinery (*Figure 4D*).

## Mutations that promote opening of the export gate partially compensate for incorrect positioning of the N-terminal export signal

The accruing data indicated that subunit docking at FlhB$_C$ might correctly position the hydrophobic N-terminal signal for recognition by the export machinery. To model the position of FlhB$_C$ relative to other components of the export machinery, we docked the structures of FlhB$_C$ and the FliPQR-FlhB$_N$ export gate into the tomographic reconstruction of the *Salmonella* SPI-1 type III secretion system (*Figure 6B*; *Meshcheryakov et al., 2013*). The model indicated that FliPQR-FlhB$_N$ and the subunit docking site on FlhB$_C$ are separated by a minimum distance of ~78 Å, and that FlhB$_C$ is positioned no more than ~22–45 Å from FlhA (*Figure 6A*; *Butan et al., 2019*). Taking FlgD as a model early flagellar subunit, the distance between the FlgD N-terminal hydrophobic signal and the GRM was found to be in the range of ~45 Å (α-helix) to ~105 Å (unfolded contour length), depending on the predicted structure adopted by the subunit N-terminus (*Figure 6A*). Based on these estimates, it seemed feasible that the hydrophobic N-terminal signal of a subunit docked at FlhB$_C$ could contact either FlhA or the FliPQR-FlhB$_N$ complex, and that this interaction might trigger opening of the export gate (*Kuhlen et al., 2018*; *Meshcheryakov et al., 2013*; *Bryant and Fraser, 2021*; *Vonderviszt et al., 1992*). If this were true, mutations that promote the open conformation of the export gate might compensate for the incorrect positioning of the hydrophobic N-terminal export signal in 'short' rod/hook subunits. One such export gate mutation, FliP-M$_{210}$A, has been shown to increase ion conductance across the bacterial inner membrane, indicating that this gate variant fails to close efficiently (*Kuhlen et al., 2020*).

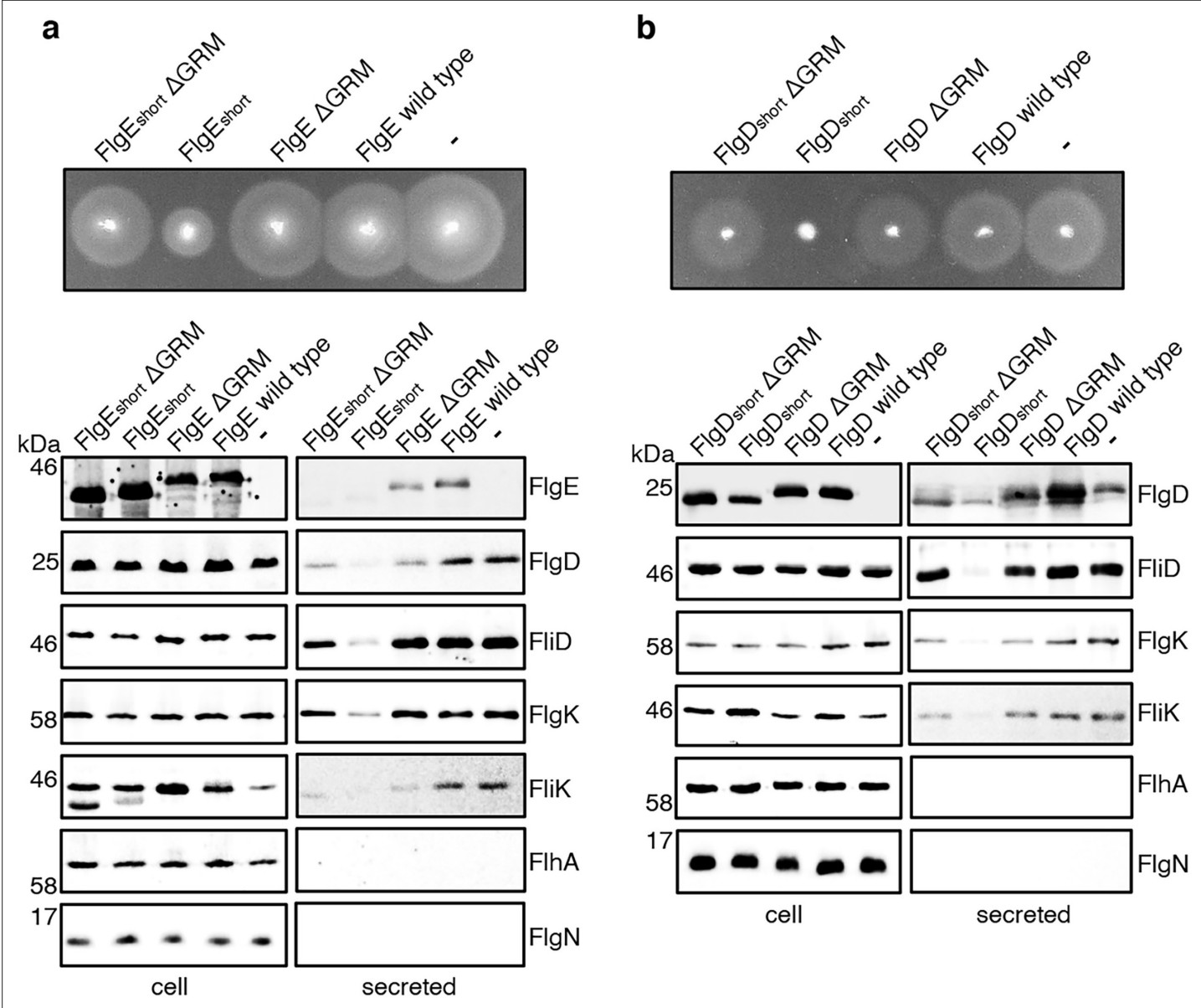

**Figure 5.** Effect on subunit export of overexpressed FlgE_short, FlgD_short and variants.

(a) Swimming motility of a *Salmonella* Δ*recA* strain expressing plasmid-encoded wild type FlgE (labelled as FlgE wild type), a FlgE variant in which residues 9–32 were deleted (labelled as FlgE_short), a FlgE variant in which residues 9–32 and residues 39–43 (corresponding to the gate-recognition motif) were deleted (labelled as FlgE_shortΔGRM), a FlgE variant in which residues 39–43 were deleted (labelled as FlgEΔGRM) or empty pTrc99a vector (labelled as -). All FlgE variants were engineered to contain an internal 3xFLAG tag for immunodetection. Motility was assessed in 0.25% soft-tryptone agar containing 100 µg/ml ampicillin and 100 µM IPTG and incubated for 4–6 hr at 37 °C (top panel). Whole cell (cell) and secreted proteins (secreted) from late-exponential-phase cultures were separated by SDS (15%)-PAGE and analysed by immunoblotting with anti-FLAG monoclonal antisera (to detect the flag tagged hook subunit FlgE) or anti-FlgD (hook cap subunit), anti-FliD (filament cap subunit), anti-FlgK (hook-filament junction subunit), anti-FliK (hook-ruler subunit), anti-FlhA (component of the export machinery) and anti-FlgN (chaperone for FlgK and FlgL) polyclonal antisera (bottom). Apparent molecular weights are in kilodaltons (kDa).(b) Swimming motility of a *Salmonella* Δ*recA* strain expressing plasmid-encoded wild type FlgD (labelled as FlgD wild type), a FlgD variant in which residues 9–32 were replaced with two repeats of the six amino acid sequence Gly-Ser-Thr-Asn-Ala-Ser (labelled as FlgD_short), a FlgD variant in which residues 9–32 were replaced with two repeats of the six amino acid sequence Gly-Ser-Thr-Asn-Ala-Ser and residues 36–40 were deleted (labelled FlgD_shortΔGRM), a FlgD variant in which residues 36–40 were deleted (labelled FlgDΔGRM) or empty pTrc99a vector (labelled as -). Motility was assessed in 0.25% soft-tryptone agar containing 100 µg/ml ampicillin and 100 µM IPTG and incubated for 4–6 hours at 37 °C (top panel). Whole cell (cell) and secreted proteins (secreted) from late-exponential-phase cultures were separated by SDS (15%)-PAGE and analysed by immunoblotting with anti-FlgD (hook cap subunit), anti-FliD (filament cap subunit), anti-FlgK (hook-filament junction subunit), anti-FliK (hook ruler subunit), anti-FlhA (component of export machinery), and anti-FlgN (chaperone for FlgK and FlgL) polyclonal antisera (bottom). Apparent molecular weights are in kilodaltons (kDa).

*Figure 5 continued on next page*

*Figure 5 continued*

The online version of this article includes the following source data for figure 5:

**Source data 1.** Full-length protein western blot of cellular proteins relating to *Figure 4A* (low contrast, anti-FLAG).

**Source data 2.** Full-length protein western blot of secreted proteins relating to *Figure 4A* (high contrast, anti-FLAG).

**Source data 3.** Full-length protein western blot of cellular and secreted proteins relating to *Figure 4A* (anti-FliD (top) and anti-FlgD (bottom)).

**Source data 4.** Full-length protein western blot of cellular and secreted proteins relating to *Figure 4A* (anti-FlgK).

**Source data 5.** Full-length protein western blot of cellular proteins relating to *Figure 4A* (anti-FliK, left).

**Source data 6.** Full-length protein western blot of secreted proteins relating to *Figure 4A* (anti-FliK, bottom right).

**Source data 7.** Full-length protein western blot of cellular and secreted proteins relating to *Figure 4A* (anti-FlhA, top).

**Source data 8.** Full-length protein western blot of cellular and secreted proteins relating to *Figure 4A* (anti-FlgN, bottom).

**Source data 9.** Full-length protein western blot of cellular proteins relating to *Figure 4B* (anti-FlgD, left).

**Source data 10.** Full-length protein western blot of secreted proteins relating to *Figure 4B* (anti-FlgD, bottom right).

**Source data 11.** Full-length protein western blot of cellular and secreted proteins relating to *Figure 4B* (anti-FliD).

**Source data 12.** Full-length protein western blot of cellular and secreted proteins relating to *Figure 4B* (anti-FlgK, top).

**Source data 13.** Full-length protein western blot of cellular and secreted proteins relating to *Figure 4B* (anti-FliK).

**Source data 14.** Full-length protein western blot of cellular and secreted proteins relating to *Figure 4B* (anti-FlhA (top) and anti-FlgN (bottom)).

To test whether the FliP-M$_{210}$A variant gate could promote export of 'short' subunits, in which the distance between the hydrophobic N-terminal signal and the GRM was reduced, a recombinant expression vector encoding FlgD$_{short}$ was introduced into *Salmonella flgD* null strains in which the *fliP* gene had been replaced with recombinant genes encoding either a functional FliP variant with an internal HA-tag (designated wild-type gate) or the equivalent HA-tagged FliP-M$_{210}$A variant (designated M$_{210}$A gate; *Figure 6C*, *Figure 6—figure supplement 1*). The swimming motility of these strains was found to be consistently stronger in the strain producing the M$_{210}$A gate compared to the strain with the wild-type gate, with the motility halo of the *fliP*-M210A-Δ*flgD* strain expressing FlgD$_{short}$ having a 50% greater diameter than that of the wild type *fliP*-Δ*flgD* strain expressing FlgD$_{short}$ (*Figure 6D*, *Figure 6—figure supplement 1*). This increase in motility indicated that the defect caused by incorrect positioning of the hydrophobic N-terminal signal relative to the GRM in FlgD$_{short}$ could indeed be partially compensated by promoting the gate open conformation.

## Discussion

T3SS substrates contain N-terminal export signals, but these have not been fully defined and how they promote subunit export remains unclear. Here, we characterised a new hydrophobic N-terminal export signal in early flagellar rod/hook subunits and showed that the position of this signal relative to the known subunit gate recognition motif (GRM) is key to subunit export.

Loss of the hydrophobic N-terminal signal in the hook cap subunit FlgD had a stronger negative effect on subunit export than deletion of the GRM that enables subunit docking at FlhB$_C$, suggesting that the hydrophobic N-terminal signal may be required to trigger an essential export step. A suppressor screen showed that the export defect caused by deleting the hydrophobic N-terminal signal could be overcome by mutations that either reintroduced small non-polar amino acids positioned 3–7 residues from the subunit N-terminus (e.g. M$_7$I), or introduced additional residues between V$_{15}$ and the GRM. In such 'gain of function' strains containing insertions, changing V$_{15}$ to alanine abolished subunit export, which was rescued by re-introduction of small non-polar residues close to the N-terminus. These data point to an essential export function for small non-polar residues close to the N-terminus of rod/hook subunits.

It was fortuitous that we chose FlgD as the model for early flagellar subunit. All early subunits contain small hydrophobic residues close to the N-terminus, but FlgD is unique in that only four (I$_3$, A$_4$, V$_5$ and V$_{15}$) of its first 25 residues are small and non-polar (*Figure 1—figure supplement 1*). Indeed, there are only three other small hydrophobic residues between the FlgD N-terminus and the GRM (*Figure 1*, *Figure 1—figure supplement 1*, *Figure 6—figure supplement 2*). While deletion of residues 2–5 in FlgD removes the critical hydrophobic N-terminal signal, similar deletions in the N-terminal regions of other rod/hook subunits reposition existing small non-polar residues close to the

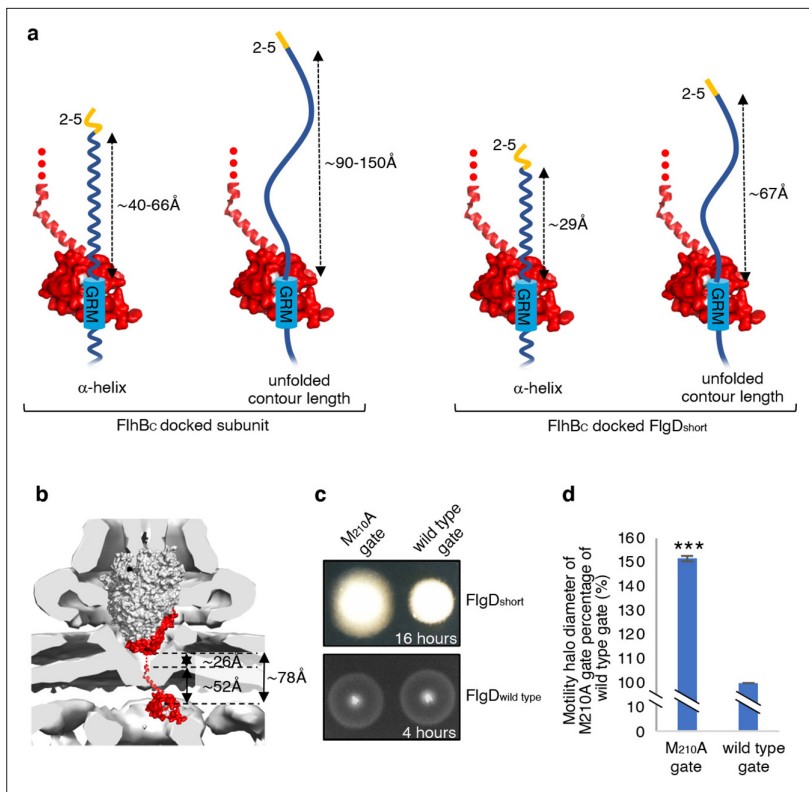

**Figure 6.** Suppression of the FlgD$_{short}$ motility defect by mutations in FliP. (**a**) A model depicting subunits docked via their gate-recognition motif (GRM, blue) at the subunit binding pocket on FlhB$_C$ (PDB: 3B0Z **Meshcheryakov et al., 2013**, red) with N-termini of early flagellar subunits adopting either an α-helical conformation separating the N-terminal hydrophobic signal (2–5, orange) and gate-recognition motif (GRM, blue) by ~40–60 ångstrom (where each amino acid is on average separated by ~1.5 Å, left) or an unfolded conformation where the unfolded contour length separating the N-terminal hydrophobic signal (2-5) and gate-recognition motif (GRM) is ~90–150 ångstrom (where each amino acid is on average separated by ~3.5 Å, middle left). Values corresponding to the distance separating the N-terminal hydrophobic signal (2–5, orange) and gate-recognition motif (GRM, blue) of a FlgD subunit variant in which residues 9–32 are replaced with two repeats of the six amino acid sequence Gly-Ser-Thr-Asn-Ala-Ser (FlgD$_{short}$) indicate that the N-terminal hydrophobic signal (2–5, orange) and gate-recognition motif (GRM, blue) are separated by ~29 ångstrom (α-helical conformation, middle right) or ~67 ångstrom (unfolded contour length, right). (**b**) Placement of the crystal structure of FlhBc (PDB:3BOZ **Kuhlen et al., 2020**; red) and the cryo-EM structure of FliPQR-FlhB (PDB:6S3L **Ward et al., 2018**) in a tomographic reconstruction of the *Salmonella* SPI-1 injectisome (EMD-8544 [60]; grey). The minimum distance between the subunit gate-recognition motif binding site on FlhB$_C$ (grey) to FlhB$_N$ (defined as *Salmonella* FlhB residue 211 [61]; ~ 78 Å) was estimated by combining: the value corresponding to the distance between the subunit binding pocket on FlhB$_C$ (**Evans et al., 2013**) (grey) and the N-terminal visible residue (D$_{229}$) in the FlhB$_C$ structure (PDB:3BOZ **Kuhlen et al., 2020**; ~ 52 Å) with the value corresponding to the minimum distance between FlhB residues 211 and 228 (based on a linear α-helical conformation; ~ 26 Å). (**c**) Swimming motility of recombinant *Salmonella flgD* null strains producing a chromosomally-encoded FliP-M$_{210}$A variant (M$_{210}$A gate, left) or wild type FliP (wild-type gate, right). Wildtype FliP and FliP-M$_{210}$A were engineered to contain an internal HA tag positioned between residue 21 and 22 to allow immunodetection of FliP. Both strains produced either a pTrc99a plasmid-encoded FlgD subunit variant in which residues 9–32 were replaced with two repeats of the six amino acid sequence Gly-Ser-Thr-Asn-Ala-Ser (FlgD$_{short}$; top panel) or a pTrc99a plasmid-encoded wild-type FlgD subunit (FlgD$_{wild type}$; bottom panel). Motility was assessed in 0.25% soft-tryptone agar containing 100 µg/ml ampicillin and 50 µM IPTG and incubated for 16 hr (top panel) or 4–6 hr at 37 °C (bottom panel). (**d**). The mean motility halo diameter of recombinant *Salmonella flgD* null strains producing a chromosomally-encoded FliP-M$_{210}$A variant (M$_{210}$A gate, left) or wild-type FliP (wild-type gate, right). Wild-type FliP and FliP-M$_{210}$A were engineered to contain an internal HA tag positioned between residue 21 and 22 to allow immunodetection of FliP. Both strains produced a pTrc99a plasmid-encoded FlgD subunit variant in which residues 9–32 were replaced with two repeats of the six amino acid sequence Gly-Ser-Thr-Asn-Ala-Ser (FlgD$_{short}$). Error bars represent the standard error of the mean calculated from at least three biological replicates. *** indicates a p-value < 0.001.

*Figure 6 continued on next page*

*Figure 6 continued*

The online version of this article includes the following source data and figure supplement(s) for figure 6:

**Figure supplement 1.** Mutations in FliP suppress the motility defect associated with FlgDshort but do not change the motility of strains producing wild type FlgD.

**Figure supplement 1—source data 1.** Full-length protein western blot of cellular proteins relating to *Figure 6— figure supplement 1A* (anti-HA).

**Figure supplement 1—source data 2.** Full-length protein western blot of cellular proteins relating to *Figure 6— figure supplement 1A* (anti-FlgD).

**Figure supplement 2** Amino acid sequence alignments of N-terminal regions of *Salmonella* FlgD cap, FlgE hook and FliK ruler with flagellar subunits from other bacterial species.

**Figure supplement 3.** The substrate binding cytoplasmic domain of FlhB is predicted to be positioned between the plane of the inner membrane and cytoplasmic domain of FlhA.

**Figure supplement 4.** A schematic of an early flagellar subunit containing an N-terminal export signal (yellow) and gate-recognition motif (GRM, blue).

N-terminus (*Figure 1—figure supplement 1*, *Figure 6—figure supplement 2*). This is perhaps why previous deletion studies in early flagellar subunits have failed to identify the hydrophobic N-terminal signal (*Evans et al., 2013*; *Hirano et al., 2005*; *Minamino et al., 1999*).

The finding that subunits lacking the hydrophobic N-terminal export signal, but not the GRM, stalled during export suggested that these two signals were recognised by the flagellar export machinery in a specific order. Mutant variants of other flagellar export substrates or export components have been observed to block the export pathway. For example, a FlgN chaperone variant lacking the C-terminal 20 residues stalls at the FliI ATPase (*Thomas et al., 2004*), while a GST-tagged FliJ binds FlhA but is unable to associate correctly with FliI so blocking wild type FliJ interaction with FlhA (*Ibuki et al., 2012*). These attenuations can be reversed by further mutations that disrupt the stalling interactions. This was also observed for FlgDΔ2–5. Loss of the hydrophobic N-terminal signal resulted in a dominant-negative effect on motility and flagellar export, but this was abolished by subsequent deletion of the GRM. This indicates that FlgDΔ2–5 stalls in the export pathway at FlhB$_C$, blocking the binding site for early flagellar subunits. These data are consistent with sequential recognition of the two export signals: the GRM first docking subunits at FlhB$_C$, and positioning the hydrophobic N-terminal signal to trigger the next export step.

The position of the subunit hydrophobic N-terminal export signal relative to the GRM appears critical for export. Engineering of *flgD* to encode variants in which the region between the N-terminus and the GRM was replaced with polypeptide sequences of varying lengths showed that these signals must be separated by a minimum of 19 residues for detectable export, with substantial export requiring separation by 30 residues (*Figure 3*). When subunits dock at FlhB$_C$, which is likely situated within or just below the plane of the inner membrane, the hydrophobic N-terminal signal is positioned close to the FlhAB-FliPQR export gate (*Figure 6B*). Subunits in which the GRM and N-terminal signal are brought closer together stall at FlhB$_C$, suggesting that the hydrophobic N-terminal signal is unable to contact its recognition site on the export machinery (*Figures 5, 6A and B*). In all flagellar rod/hook subunits, the GRM is positioned at least 30 residues from the N-terminus (*Figure 1—figure supplement 1*, *Figure 6—figure supplement 2*). The physical distance between the two signals will depend on the structure adopted by the subunit N-terminus (*Figure 6A*). The N-terminal region of flagellar subunits is often unstructured in solution (*Kuwajima et al., 1989*; *Minamino and Macnab, 1999*; *Kornacker and Newton, 1994*; *Evans et al., 2013*; *Végh et al., 2006*; *Vonderviszt et al., 1992*), and such disorder may be an intrinsic feature of flagellar export signals (*Kuwajima et al., 1989*; *Minamino and Macnab, 1999*; *Kornacker and Newton, 1994*; *Evans et al., 2013*; *Végh et al., 2006*; *Weber-Sparenberg et al., 2006*; *Aizawa et al., 1990*), as they are typical in other bacterial export N-terminal substrate signals such as those of the Sec and Tat systems (*Tsirigotaki et al., 2017*; *Palmer and Berks, 2012*). Unstructured signals may facilitate multiple interactions with different binding partners during export, and in the case of export systems that transport unfolded proteins they may aid initial entry of substrates into narrow export channels (*Tsirigotaki et al., 2017*).

As yet, nothing is known about the structure of the subunit N-terminal domain upon interaction with the flagellar export machinery. Signal peptides in TAT pathway substrates switch between disordered

and α-helical conformations depending on the hydrophobicity of the environment (**San Miguel et al., 2003**). It therefore seems likely that local environments along the flagellar export pathway will influence the conformation of subunit export signals (**Kuhlen et al., 2018**; **Erhardt et al., 2017**). Interestingly, FlgD variants that contain additional sequence that position the N-terminal export signal and GRM further apart far above the required threshold did not impede export, which argues that the sequence between both export signals is unfolded (**Figure 2**).

If the region between the hydrophobic N-terminal signal and the GRM is unstructured and extended, this would correspond to a polypeptide contour length of approximately 72–105 Å (where the length of one amino acid is ~3.6 Å). If the same region were to fold as an α-helix, its length would be approximately 30–36 Å (where one amino acid rises every ~1.5 Å). Placement of the AlphaFold predicted structure of full length FlhB into a tomogram of the T3SS suggests that FlhB$_C$ is positioned below the plane of the inner membrane but above the nonameric ring of FlhA$_C$ (**Figure 6—figure supplement 3**). Without further structural information on subunit interactions with the flagellar export machinery and the precise position of FlhB$_C$ within the machinery, it is difficult to determine precisely where the hydrophobic N-terminal signal contacts the machinery.

We speculate that one function of the subunit hydrophobic N-terminal signal might be to trigger opening of the FlhAB-FliPQR export gate, which rests in an energetically favourable closed conformation to maintain the permeability barrier across the bacterial inner membrane (**Kuhlen et al., 2018**; **Butan et al., 2019**; **Ward et al., 2018**; **Kuhlen et al., 2020**; **Bryant and Fraser, 2021**). The atomic resolution structure of FliPQR showed that it contains three gating regions (**Kuhlen et al., 2018**). FliR provides a loop (the R-plug) that sits within the core of the structure. Below this, five copies of FliP each provides three methionine residues that together form a methionine-rich ring (M-gasket) under which ionic interactions between adjacent FliQ subunits hold the base of the structure shut (Q-latch). Mutational and evolutionary analyses have shown that the R-plug, M-gasket, and Q-latch stabilize the closed export gate conformation to maintain the membrane barrier, preventing the leakage of small molecules whilst allowing the passage of substrates into the export channel (**Bryant and Fraser, 2021**; **Hüsing et al., 2021**). We have also shown that the FliPQR export gate opens and closes in response to export substrate availability, indicating that the export gate reverts to a closed conformation in the absence of export substrates, thereby maintaining the integrity of the cell membrane (**Bryant and Fraser, 2021**). These data indicate that the export gate must be opened in response to substrate docking at the export machinery (**Bryant and Fraser, 2021**; **Hüsing et al., 2021**).

If the function of the subunit hydrophobic N-terminal signal was to trigger opening of this gate, we hypothesised that mutations which destabilised the gate's closed conformation would suppress the motility defect associated with FlgD$_{short}$, in which the distance between the N-terminal signal and the GRM is reduced. Introduction of the FliP-M$_{210}$A mutation, which partially destabilises the gate's closed state, did indeed partly suppress the FlgD$_{short}$ motility defect. We did not find export gate variants that completely destabilised gate closure, but it may be that such mutations disrupt the membrane permeability barrier (**Ward et al., 2018**). This could also explain why in screens for suppressors of FlgDΔ2–5 or FlgD$_{short}$ we did not isolate mutations in genes encoding export gate components (data not shown).

The surface-exposed hydrophobic GRM-binding pocket on FlhB$_C$ is well conserved across the T3SS SctU family, to which FlhB belongs (**Evans et al., 2013**; **Zarivach et al., 2008**; **Lountos et al., 2009**). Furthermore, the GRM is conserved in all four injectisome early subunits (SctI, SctF, SctP, OrgC) and is located at least 30 residues away from small hydrophobic residues near the subunit N-terminus (**Figure 6—figure supplement 4**). It therefore seems plausible that the 'dual signal' mechanism we propose for early flagellar export operates in all T3SS pathways.

In many other pathways, the presence of a substrate triggers opening or assembly of the export channel. The outer membrane chitin transporter in *Vibrio* adopts a closed conformation in which the N-terminus of a neighbouring subunit acts as a pore plug (**Aunkham et al., 2018**). Chitin binding to the transporter ejects the plug, opening the transport channel and allowing chitin transport (**Aunkham et al., 2018**). In the Sec pathway, interactions of SecA, ribosomes or pre-proteins with SecYEG can induce conformational changes that promote channel opening (**Tsirigotaki et al., 2017**; **Ge et al., 2014**; **Zimmer et al., 2008**; **Voorhees et al., 2014**). In the TAT system, which transports folded substrates across the cytoplasmic membrane, substrate binding to the TatBC complex triggers association with, and subsequent polymerisation of, TatA, which is required for substrate translocation (**Palmer and Berks, 2012**; **Mori and Cline, 2002**). All of these mechanisms serve both to

conserve energy and prevent disruption of the membrane permeability barrier. Our data suggest that in a comparable way the signal of non-polar residues within the N-termini of early rod/hook subunits trigger export gate opening.

# Materials and methods

## Key resources table

| Reagent type (species) or resource | Designation | Source or reference | Identifiers | Additional information |
|---|---|---|---|---|
| Strain, strain background, (*Salmonella enterica* serovar Typhimurium) | SJW1103 | doi:10.1099/00221287-130-12-3339 | wildtype | This strain can be obtained from the Fraser lab upon request |
| Strain, strain background, (*Salmonella enterica* serovar Typhimurium) | *recA* null | This work | *recA* gene replaced with kanamycin resistance cassette | This strain can be obtained from the Fraser lab upon request |
| Strain, strain background, (*Salmonella enterica* serovar Typhimurium) | *flgD* null | doi:10.1038/nature12682 | *flgD* gene replaced with kanamycin resistance cassette | This strain can be obtained from the Fraser lab upon request |
| Strain, strain background, (*Salmonella enterica* serovar Typhimurium) | *fliP*(M210A)internal HA tag, *flgD* null | This work | Triple HA tag inserted between residue 21 and 22 of *fliP* and M210A mutation introduced into the *fliP* gene, *flgD* gene replaced with kanamycin resistance cassette. | This strain can be obtained from the Fraser lab upon request |
| Strain, strain background, (*Salmonella enterica* serovar Typhimurium) | *fliP* internal HA tag, *flgD* null | This work | Triple HA tag inserted between residue 21 and 22 introduced into the *fliP* gene, *flgD* gene replaced with kanamycin resistance cassette. | This strain can be obtained from the Fraser lab upon request |
| Recombinant DNA reagent | pTrc99a FlgD | This work | FlgD residues 1-232aa | This vector can be obtained from the Fraser lab upon request |
| Recombinant DNA reagent | pTrc99a FlgDΔ2–5 | This work | FlgD residues 1, 6-232aa | This vector can be obtained from the Fraser lab upon request |
| Recombinant DNA reagent | pTrc99a FlgDΔ6–10 | This work | FlgD residues 1–5, 11-232aa | This vector can be obtained from the Fraser lab upon request |
| Recombinant DNA reagent | pTrc99a FlgDΔ11–15 | This work | FlgD residues 1–10, 16-232aa | This vector can be obtained from the Fraser lab upon request |
| Recombinant DNA reagent | pTrc99a FlgDΔ16–20 | This work | FlgD residues 1–15, 21-232aa | This vector can be obtained from the Fraser lab upon request |
| Recombinant DNA reagent | pTrc99a FlgDΔ21–25 | This work | FlgD residues 1–20, 26-232aa | This vector can be obtained from the Fraser lab upon request |
| Recombinant DNA reagent | pTrc99a FlgDΔ26–30 | This work | FlgD residues 1–25, 31-232aa | This vector can be obtained from the Fraser lab upon request |
| Recombinant DNA reagent | pTrc99a FlgDΔ31–35 | This work | FlgD residues 1–30, 36-232aa | This vector can be obtained from the Fraser lab upon request |
| Recombinant DNA reagent | pTrc99a FlgDΔ36–40 | This work | FlgD residues 1–35, 41-232aa | This vector can be obtained from the Fraser lab upon request |
| Recombinant DNA reagent | pTrc99a FlgDΔ41–45 | This work | FlgD residues 1–40, 46-232aa | This vector can be obtained from the Fraser lab upon request |
| Recombinant DNA reagent | pTrc99a FlgDΔ46–50 | This work | FlgD residues 1–45, 51-232aa | This vector can be obtained from the Fraser lab upon request |
| Recombinant DNA reagent | pTrc99a FlgDΔ2–5, ΔGRM | This work | FlgD residues 1, 6–35, 41-232aa | This vector can be obtained from the Fraser lab upon request |
| Recombinant DNA reagent | pTrc99a FlgDΔ2–5 19(AGAGAG)20 | This work | FlgD residues 1–19, Ala-Gly-Ala-Gly-Ala-Gly, 20-232aa | This vector can be obtained from the Fraser lab upon request |

*Continued on next page*

*Continued*

| Reagent type (species) or resource | Designation | Source or reference | Identifiers | Additional information |
|---|---|---|---|---|
| Recombinant DNA reagent | pTrc99a FlgDΔ2–5 19(STSTST)20 | This work | FlgD residues 1–19, Ser-Thr-Ser-Thr-Ser-Thr, 20-232aa | This vector can be obtained from the Fraser lab upon request |
| Recombinant DNA reagent | pTrc99a FlgDΔ2–5 19(GSGSMT)20 | This work | FlgD residues 1–19, Gly-Ser-Gly-Ser-Met-Thr, 20-232aa | This vector can be obtained from the Fraser lab upon request |
| Recombinant DNA reagent | pTrc99a FlgDΔ9–32, 4xRpt | This work | FlgD residues 1–8, 4 x(Gly-Ser-Thr-Asn-Ala-Ser), 33-232aa | This vector can be obtained from the Fraser lab upon request |
| Recombinant DNA reagent | pTrc99a FlgDΔ9–32, 3xRpt | This work | FlgD residues 1–8, 3 x(Gly-Ser-Thr-Asn-Ala-Ser), 33-232aa | This vector can be obtained from the Fraser lab upon request |
| Recombinant DNA reagent | pTrc99a FlgDΔ9–32, 2xRpt (FlgDshort) | This work | FlgD residues 1–8, 2 x(Gly-Ser-Thr-Asn-Ala-Ser), 33-232aa | This vector can be obtained from the Fraser lab upon request |
| Recombinant DNA reagent | pTrc99a FlgDΔ9–32, 1xRpt | This work | FlgD residues 1–8, Gly-Ser-Thr-Asn-Ala-Ser, 33-232aa | This vector can be obtained from the Fraser lab upon request |
| Recombinant DNA reagent | pTrc99a FlgDΔ9–32 | This work | FlgD residues 1–8, 33-232aa | This vector can be obtained from the Fraser lab upon request |
| Recombinant DNA reagent | pTrc99a FlgDΔ9–32, 2xRpt + 1 | This work | FlgD residues 1–8, Gly, 2 x(Gly-Ser-Thr-Asn-Ala-Ser), 33-232aa | This vector can be obtained from the Fraser lab upon request |
| Recombinant DNA reagent | pTrc99a FlgDΔ9–32, 2xRpt + 2 | This work | FlgD residues 1–8, Gly-Ser, 2 x(Gly-Ser-Thr-Asn-Ala-Ser), 33-232aa | This vector can be obtained from the Fraser lab upon request |
| Recombinant DNA reagent | pTrc99a FlgDΔ9–32, 2xRpt + 3 | This work | FlgD residues 1–8, Gly-Ser-Thr, 2 x(Gly-Ser-Thr-Asn-Ala-Ser), 33-232aa | This vector can be obtained from the Fraser lab upon request |
| Recombinant DNA reagent | pTrc99a FlgDΔ9–32, 2xRpt + 4 | This work | FlgD residues 1–8, Gly-Ser-Thr-Asn, 2 x(Gly-Ser-Thr-Asn-Ala-Ser), 33-232aa | This vector can be obtained from the Fraser lab upon request |
| Recombinant DNA reagent | pTrc99a FlgDΔ9–32, 2xRpt + 5 | This work | FlgD residues 1–8, Gly-Ser-Thr-Asn-Ala, 2 x(Gly-Ser-Thr-Asn-Ala-Ser), 33-232aa | This vector can be obtained from the Fraser lab upon request |
| Recombinant DNA reagent | pTrc99a FlgDshort, ΔGRM | This work | FlgD residues 1–8, 2 x(Gly-Ser-Thr-Asn-Ala-Ser), 33–35, 41-232aa | This vector can be obtained from the Fraser lab upon request |
| Recombinant DNA reagent | pTrc99a FlgDΔ2–5 | This work | FlgD residues 1, 6-232aa | This vector can be obtained from the Fraser lab upon request |
| Recombinant DNA reagent | pTrc99a FlgDΔ2–5, M7I | This work | FlgD residues 1, 6-232aa, M7I | This vector can be obtained from the Fraser lab upon request |
| Recombinant DNA reagent | pTrc99a FlgDΔ2–5, M7V | This work | FlgD residues 1, 6-232aa, M7V | This vector can be obtained from the Fraser lab upon request |
| Recombinant DNA reagent | pTrc99a FlgDΔ2–5, N8I | This work | FlgD residues 1, 6-232aa, N8I | This vector can be obtained from the Fraser lab upon request |
| Recombinant DNA reagent | pTrc99a FlgDΔ2–5, D9A | This work | FlgD residues 1, 6-232aa, D9A | This vector can be obtained from the Fraser lab upon request |
| Recombinant DNA reagent | pTrc99a FlgDΔ2–5, P10L | This work | FlgD residues 1, 6-232aa, P10L | This vector can be obtained from the Fraser lab upon request |
| Recombinant DNA reagent | pTrc99a FlgDΔ2–5, T11I | This work | FlgD residues 1, 6-232aa, T11I | This vector can be obtained from the Fraser lab upon request |
| Recombinant DNA reagent | pTrc99a FlgDΔ2–5, 23(TTGSGS)24 | This work | FlgD residues 1–23, Thr-Thr-Gly-Ser-Gly-Ser, 24-232aa | This vector can be obtained from the Fraser lab upon request |
| Recombinant DNA reagent | pTrc99a FlgDΔ2–5, 23(TTGSGSTTGSGS)24 | This work | FlgD residues 1–23, Thr-Thr-Gly-Ser-Gly-Ser-Thr-Thr-Gly-Ser-Gly-Ser, 24-232aa | This vector can be obtained from the Fraser lab upon request |
| Recombinant DNA reagent | pTrc99a FlgDΔ2–5, 27(GSMTGS)28 | This work | FlgD residues 1–27, Gly-Ser-Met-Thr-Gly-Ser, 28-232aa | This vector can be obtained from the Fraser lab upon request |

*Continued on next page*

*Continued*

| Reagent type (species) or resource | Designation | Source or reference | Identifiers | Additional information |
|---|---|---|---|---|
| Recombinant DNA reagent | pTrc99a FlgDΔ9–32, 8 (2xGSTNAS)33, V15A | This work | FlgD residues 1–8, 2 x(Gly-Ser-Thr-Asn-Ala-Ser), 33-232aa, V15A | This vector can be obtained from the Fraser lab upon request |
| Recombinant DNA reagent | pTrc99a FlgDΔ9–32, 8 (2xGSTNAS)33, V15A, M7I | This work | FlgD residues 1–8, 2 x(Gly-Ser-Thr-Asn-Ala-Ser), 33-232aa, M7I | This vector can be obtained from the Fraser lab upon request |
| Recombinant DNA reagent | pTrc99a FlgDΔ9–32, 8 (2xGSTNAS)33, V15A, D9A | This work | FlgD residues 1–8, 2 x(Gly-Ser-Thr-Asn-Ala-Ser), 33-232aa, D9A | This vector can be obtained from the Fraser lab upon request |
| Recombinant DNA reagent | pTrc99a FlgDΔ9–32, 8 (2xGSTNAS)33, V15A, T11I | This work | FlgD residues 1–8, 2 x(Gly-Ser-Thr-Asn-Ala-Ser), 33-232aa, T11I | This vector can be obtained from the Fraser lab upon request |
| Recombinant DNA reagent | pTrc99a FlgDΔ9–32, 8 (2xGSTNAS)33, V15A, G14V | This work | FlgD residues 1–8, 2 x(Gly-Ser-Thr-Asn-Ala-Ser), 33-232aa, G14V | This vector can be obtained from the Fraser lab upon request |
| Recombinant DNA reagent | pTrc99a FlgDΔ9–32, 8 (2xGSTNAS-TNPGSTNAS)33 | This work | FlgD residues 1–8, 2 x(Gly-Ser-Thr-Asn-Ala-Ser), (Thr-Asn-Pro-Gly-Ser-Thr-Asn-Ala-Ser) 33-232aa, | This vector can be obtained from the Fraser lab upon request |
| Recombinant DNA reagent | pTrc99a FlgDΔ9–32, 8 (2xGSTNAS-GNASGSTNAS)33 | This work | FlgD residues 1–8, 2 x(Gly-Ser-Thr-Asn-Ala-Ser), (Gly-Asn-Ala-Ser-Gly-Ser-Thr-Asn-Ala-Ser) 33-232aa, | This vector can be obtained from the Fraser lab upon request |
| Recombinant DNA reagent | pTrc99a FlgDΔ9–32, 8 (2xGSTNAS-QSSFLTL LVAQLKNQDPTNPLQNNELTTQLA)33 | This work | FlgD residues 1–8, 2 x(Gly-Ser-Thr-Asn-Ala-Ser), (Gln-Ser-Ser-Phe-Leu-Thr-Leu-Leu-Val-Ala-Gln-Leu-Lys-Asn-Gln-Asp-Pro-Thr-Asn-Pro-Leu-Asn-Asn-Glu-Leu-Thr-Thr-Gln-Leu-Ala), 33-232aa, | This vector can be obtained from the Fraser lab upon request |
| Recombinant DNA reagent | pTrc99a FlgDΔ9–32, 8 (2xGSTNAS-TNASGSTNAS)33 | This work | FlgD residues 1–8, 2 x(Gly-Ser-Thr-Asn-Ala-Ser), (Thr-Asn-Ala-Ser-Gly-Ser-Thr-Asn-Ala-Ser) 33-232aa, | This vector can be obtained from the Fraser lab upon request |
| Recombinant DNA reagent | pTrc99a FlgDΔ9–32, 8 (2xGSTNAS-QSSLGSTNAS)34 | This work | FlgD residues 1–8, 2 x(Gly-Ser-Thr-Asn-Ala-Ser), (Gln-Ser-Ser-Leu-Gly-Ser-Thr-Asn-Ala-Ser) 33-232aa, | This vector can be obtained from the Fraser lab upon request |
| Recombinant DNA reagent | pTrc99a FlgDΔ9–32, 8 (2xGSTNAS-QNASGSTNAS)35 | This work | FlgD residues 1–8, 2 x(Gly-Ser-Thr-Asn-Ala-Ser), (Gln-Asn-Ala-Ser-Gly-Ser-Thr-Asn-Ala-Ser) 33-232aa, | This vector can be obtained from the Fraser lab upon request |
| Recombinant DNA reagent | pTrc99a FlgDΔ9–32, 8 (2xGSTNAS-TNTFGTLIAS)36 | This work | FlgD residues 1–8, 2 x(Gly-Ser-Thr-Asn-Ala-Ser), (Thr-Asn-Thr-Phe-Gly-Thr-Leu-Iso-Ala-Ser) 33-232aa, | This vector can be obtained from the Fraser lab upon request |
| Recombinant DNA reagent | pTrc99a FlgG | This work | FlgG residues 1–144, FLAGx3, 145-260aa | This vector can be obtained from the Fraser lab upon request |
| Recombinant DNA reagent | pTrc99a FlgGΔshort | This work | FlgG residues 1–10, 35–144, FLAGx3, 145-260aa | This vector can be obtained from the Fraser lab upon request |
| Recombinant DNA reagent | pTrc99a FlgGshort+ linker | This work | FlgG residues 1–10, 4 x(Gly-Ser-Thr-Asn-Ala-Ser) 35–144, FLAGx3, 145-260aa | This vector can be obtained from the Fraser lab upon request |
| Recombinant DNA reagent | pTrc99a FlgE | This work | FlgE residues 1–234, FLAGx3, 235-403aa | This vector can be obtained from the Fraser lab upon request |
| Recombinant DNA reagent | pTrc99a FlgEshort | This work | FlgE residues 1–8, 33–234, FLAGx3, 235-403aa | This vector can be obtained from the Fraser lab upon request |

*Continued on next page*

*Continued*

| Reagent type (species) or resource | Designation | Source or reference | Identifiers | Additional information |
|---|---|---|---|---|
| Recombinant DNA reagent | pTrc99a FlgEshort+ linker | This work | FlgE residues 1–8, 4 x(Gly-Ser-Thr-Asn-Ala-Ser) 33–234, FLAGx3, 235-403aa | This vector can be obtained from the Fraser lab upon request |
| Recombinant DNA reagent | pTrc99a FlgEΔGRM | This work | FlgE residues 1–38, 44–234, FLAGx3, 235-403aa | This vector can be obtained from the Fraser lab upon request |
| Recombinant DNA reagent | pTrc99a FlgEshort, ΔGRM | This work | FlgE residues 1–8, 33–38, 44–234, FLAGx3, 235-403aa | This vector can be obtained from the Fraser lab upon request |
| Recombinant DNA reagent | pTrc99a FliKmyc | This work | FliK residues 1-405aa, myc | This vector can be obtained from the Fraser lab upon request |
| Recombinant DNA reagent | pTrc99a FliKmycΔ2–8 | This work | FliK residues 1, 9-405aa, myc | This vector can be obtained from the Fraser lab upon request |
| Antibody | anti-FLAG (Mouse monoclonal) | Sigma-Aldrich | Cat# F3165, RRID:AB_259529 | Mouse monoclonal against FLAG tag (1:1000) |
| Antibody | Anti-HA Tag, HRP conjugate (Mouse monoclonal) | Thermo Fisher Scientific | Cat # 26183-HRP, RRID:AB_2533056 | Mouse monoclonal against HA tag (1:1000) |
| Antibody | anti-Myc (9B11), HRP conjugate (Mouse monoclonal) | Cell signalling technology | Cat # 2040, RRID:AB_2148465 | Mouse monoclonal against Myc tag (1:1000) |
| Antibody | anti-FlgD (Rabbit polyclonal) | doi:10.1038/nature12682 | | Rabbit polyclonal against *Salmonella* FlgD (1:1000). This antibody can be obtained from the Fraser lab upon request. |
| Antibody | anti-FliK (Rabbit polyclonal) | This work | | Rabbit polyclonal against *Salmonella* FliK (1:1000). This antibody can be obtained from the Fraser lab upon request. |
| Antibody | anti-FlgK (Rabbit polyclonal) | doi: 10.1111/mmi.14731 | | Rabbit polyclonal against *Salmonella* FlgK (1:1000). This antibody can be obtained from the Fraser lab upon request. |
| Antibody | anti-FlgL (Rabbit polyclonal) | This work | | Rabbit polyclonal against *Salmonella* FlgL (1:1000). This antibody can be obtained from the Fraser lab upon request. |
| Antibody | anti-FlgN (Rabbit polyclonal) | doi: 10.1111/mmi.14731 | | Rabbit polyclonal against *Salmonella* FlgN (1:1000). This antibody can be obtained from the Fraser lab upon request. |
| Antibody | anti-FlhA (Rabbit polyclonal) | doi: 10.1111/mmi.14731 | | Rabbit polyclonal against *Salmonella* FlhA (1:1000). This antibody can be obtained from the Fraser lab upon request. |

## Bacterial strains, plasmids, and growth conditions

*Salmonella* strains and plasmids used in this study are listed in *Table 1*. The Δ*flgD*::K$_m$$^R$ strain in which the *flgD* gene was replaced by a kanamycin resistance cassette was constructed using the λ Red recombinase system (*Datsenko and Wanner, 2000*). Strains containing chromosomally encoded FliP variants were constructed by aph-I-SceI Kanamycin resistance cassette replacement using pWRG730 (*Hoffmann et al., 2017*). Recombinant proteins were expressed in *Salmonella* from the isopropyl β-D-thiogalactoside-inducible (IPTG) inducible plasmid pTrc99a (*Amann et al., 1988*). Bacteria were cultured at 30–37°C in Luria-Bertani (LB) broth containing ampicillin (100 μg/ml).

## Flagellar subunit export assay

*Salmonella* strains were cultured at 37 °C in LB broth containing ampicillin and IPTG to mid-log phase (OD$_{600nm}$ 0.6–0.8). Cells were centrifuged (6000 x g, 3 min) and resuspended in fresh media and grown for a further 60 min at 37 °C. The cells were pelleted by centrifugation (16,000 x g, 5 min) and the supernatant passed through a 0.2 μm nitrocellulose filter. Proteins were precipitated with 10% trichloroacetic acid (TCA) and 1% Triton X-100 on ice for 1 hr, pelleted by centrifugation (16,000 x g,

**Table 1.** Strains and recombinant plasmids.

| Strains | Description |
| --- | --- |
| *Salmonella typhimurium* | |
| SJW1103 | wildtype |
| *recA* null | Δ*recA*::kmR |
| *flgD* null | Δ*flgD*::kmR |
| *fliP*(M$_{210}$A)$_{internalHAtag}$, *flgD* null | *fliP*(M$_{210}$A) 21 (3xHA tag)22, Δ*flgD*::kmR |
| *fliP*$_{internalHAtag}$, *flgD* null | *fliP* 21 (3xHA tag)22, Δ*flgD*::kmR |
| Plasmids | |
| pTrc99a FlgD | 1-232aa |
| pTrc99a FlgDΔ2–5 | 1, 6-232aa |
| pTrc99a FlgDΔ6–10 | 1–5, 11-232aa |
| pTrc99a FlgDΔ11–15 | 1–10, 16-232aa |
| pTrc99a FlgDΔ16–20 | 1–15, 21-232aa |
| pTrc99a FlgDΔ21–25 | 1–20, 26-232aa |
| pTrc99a FlgDΔ26–30 | 1–25, 31-232aa |
| pTrc99a FlgDΔ31–35 | 1–30, 36-232aa |
| pTrc99a FlgDΔ36–40 | 1–35, 41-232aa |
| pTrc99a FlgDΔ41–45 | 1–40, 46-232aa |
| pTrc99a FlgDΔ46–50 | 1–45, 51-232aa |
| pTrc99a FlgDΔ2–5, ΔGRM | 1, 6–35, 41-232aa |
| pTrc99a FlgDΔ2–5 $^{19}$(AGAGAG)20 | 1–19, Ala-Gly-Ala-Gly-Ala-Gly, 20-232aa |
| pTrc99a FlgDΔ2–5 $^{19}$(STSTST)20 | 1–19, Ser-Thr-Ser-Thr-Ser-Thr, 20-232aa |
| pTrc99a FlgDΔ2–5 $^{19}$(GSGSMT)20 | 1–19, Gly-Ser-Gly-Ser-Met-Thr, 20-232aa |
| pTrc99a FlgDΔ9–32, 4xRpt | 1–8, 4 x(Gly-Ser-Thr-Asn-Ala-Ser), 33-232aa |
| pTrc99a FlgDΔ9–32, 3xRpt | 1–8, 3 x(Gly-Ser-Thr-Asn-Ala-Ser), 33-232aa |
| pTrc99a FlgDΔ9–32, 2xRpt (FlgD$_{short}$) | 1–8, 2 x(Gly-Ser-Thr-Asn-Ala-Ser), 33-232aa |
| pTrc99a FlgDΔ9–32, 1xRpt | 1–8, Gly-Ser-Thr-Asn-Ala-Ser, 33-232aa |
| pTrc99a FlgDΔ9–32 | 1–8, 33-232aa |
| pTrc99a FlgDΔ9–32, 2xRpt + 1 | 1–8, Gly, 2 x(Gly-Ser-Thr-Asn-Ala-Ser), 33-232aa |
| pTrc99a FlgDΔ9–32, 2xRpt + 2 | 1–8, Gly-Ser, 2 x(Gly-Ser-Thr-Asn-Ala-Ser), 33-232aa |
| pTrc99a FlgDΔ9–32, 2xRpt + 3 | 1–8, Gly-Ser-Thr, 2 x(Gly-Ser-Thr-Asn-Ala-Ser), 33-232aa |
| pTrc99a FlgDΔ9–32, 2xRpt + 4 | 1–8, Gly-Ser-Thr-Asn, 2 x(Gly-Ser-Thr-Asn-Ala-Ser), 33-232aa |
| pTrc99a FlgDΔ9–32, 2xRpt + 5 | 1–8, Gly-Ser-Thr-Asn-Ala, 2 x(Gly-Ser-Thr-Asn-Ala-Ser), 33-232aa |
| pTrc99a FlgDshort, ΔGRM | 1–8, 2 x(Gly-Ser-Thr-Asn-Ala-Ser), 33–35, 41-232aa |
| pTrc99a FlgDΔ2–5 | 1, 6-232aa |
| pTrc99a FlgDΔ2–5, M$_7$I | 1, 6-232aa, M$_7$I |
| pTrc99a FlgDΔ2–5, M$_7$V | 1, 6-232aa, M$_7$V |
| pTrc99a FlgDΔ2–5, N$_8$I | 1, 6-232aa, N$_8$I |
| pTrc99a FlgDΔ2–5, D9A | 1, 6-232aa, D$_9$A |
| pTrc99a FlgDΔ2–5, P$_{10}$L | 1, 6-232aa, P$_{10}$L |
| pTrc99a FlgDΔ2–5, T$_{11}$I | 1, 6-232aa, T$_{11}$I |
| pTrc99a FlgDΔ2–5, $^{23}$(TTGSGS)$^{24}$ | 1–23, Thr-Thr-Gly-Ser-Gly-Ser, 24-232aa |

*Table 1 continued on next page*

*Table 1 continued*

| Strains | Description |
|---|---|
| pTrc99a FlgDΔ2–5, [23](TTGSGSTTGSGS)[24] | 1–23, Thr-Thr-Gly-Ser-Gly-Ser-Thr-Thr-Gly-Ser-Gly-Ser, 24-232aa |
| pTrc99a FlgDΔ2–5, [27](GSMTGS)[28] | 1–27, Gly-Ser-Met-Thr-Gly-Ser, 28-232aa |
| pTrc99a FlgDΔ9–32, [8](2xGSTNAS)[33], $V_{15}$A | 1–8, 2 x(Gly-Ser-Thr-Asn-Ala-Ser), 33-232aa, V15A |
| pTrc99a FlgDΔ9–32, [8](2xGSTNAS)[33], $V_{15}$A, $M_7$I | 1–8, 2 x(Gly-Ser-Thr-Asn-Ala-Ser), 33-232aa, M7I |
| pTrc99a FlgDΔ9–32, [8](2xGSTNAS)[33], $V_{15}$A, $D_9$A | 1–8, 2 x(Gly-Ser-Thr-Asn-Ala-Ser), 33-232aa, D9A |
| pTrc99a FlgDΔ9–32, [8](2xGSTNAS)[33], $V_{15}$A, $T_{11}$I | 1–8, 2 x(Gly-Ser-Thr-Asn-Ala-Ser), 33-232aa, T11I |
| pTrc99a FlgDΔ9–32, [8](2xGSTNAS)[33], $V_{15}$A, $G_{14}$V | 1–8, 2 x(Gly-Ser-Thr-Asn-Ala-Ser), 33-232aa, G14V |
| pTrc99a FlgDΔ9–32, [8](2xGSTNAS-TNPGSTNAS)[33] | 1–8, 2 x(Gly-Ser-Thr-Asn-Ala-Ser), (Thr-Asn-Pro-Gly-Ser-Thr-Asn-Ala-Ser) 33-232aa, |
| pTrc99a FlgDΔ9–32, [8](2xGSTNAS-GNASGSTNAS)[33] | 1–8, 2 x(Gly-Ser-Thr-Asn-Ala-Ser), (Gly-Asn-Ala-Ser-Gly-Ser-Thr-Asn-Ala-Ser) 33-232aa, |
| pTrc99a FlgDΔ9–32, [8](2xGSTNAS-QSSFLTLLVAQLKNQDPTNPLQNNELTTQLA)33 | 1–8, 2 x(Gly-Ser-Thr-Asn-Ala-Ser), (Gln-Ser-Ser-Phe-Leu-Thr-Leu-Leu-Val-Ala-Gln-Leu-Lys-Asn-Gln-Asp-Pro-Thr-Asn-Pro-Leu-Asn-Asn-Glu-Leu-Thr-Thr-Gln-Leu-Ala), 33-232aa, |
| pTrc99a FlgDΔ9–32, [8](2xGSTNAS-TNASGSTNAS)[33] | 1–8, 2 x(Gly-Ser-Thr-Asn-Ala-Ser), (Thr-Asn-Ala-Ser-Gly-Ser-Thr-Asn-Ala-Ser) 33-232aa, |
| pTrc99a FlgDΔ9–32, [8](2xGSTNAS-QSSLGSTNAS)[34] | 1–8, 2 x(Gly-Ser-Thr-Asn-Ala-Ser), (Gln-Ser-Ser-Leu-Gly-Ser-Thr-Asn-Ala-Ser) 33-232aa, |
| pTrc99a FlgDΔ9–32, [8](2xGSTNAS-QNASGSTNAS)[35] | 1–8, 2 x(Gly-Ser-Thr-Asn-Ala-Ser), (Gln-Asn-Ala-Ser-Gly-Ser-Thr-Asn-Ala-Ser) 33-232aa, |
| pTrc99a FlgDΔ9–32, [8](2xGSTNAS-TNTFGTLIAS)[36] | 1–8, 2 x(Gly-Ser-Thr-Asn-Ala-Ser), (Thr-Asn-Thr-Phe-Gly-Thr-Leu-Iso-Ala-Ser) 33-232aa, |
| pTrc99a FlgG | 1–144, FLAGx3, 145-260aa |
| pTrc99a FlgGΔshort | 1–10, 35–144, FLAGx3, 145-260aa |
| pTrc99a FlgGshort+ linker | 1–10, 4 x(Gly-Ser-Thr-Asn-Ala-Ser) 35–144, FLAGx3, 145-260aa |
| pTrc99a FlgE | 1–234, FLAGx3, 235-403aa |
| pTrc99a FlgEshort | 1–8, 33–234, FLAGx3, 235-403aa |
| pTrc99a FlgEshort+ linker | 1–8, 4 x(Gly-Ser-Thr-Asn-Ala-Ser) 33–234, FLAGx3, 235-403aa |
| pTrc99a FlgEΔGRM | 1–38, 44–234, FLAGx3, 235-403aa |
| pTrc99a FlgEshort, ΔGRM | 1–8, 33–38, 44–234, FLAGx3, 235-403aa |
| pTrc99a FliKmyc | 1-405aa, myc |
| pTrc99a FliKmycΔ2–8 | 1, 9-405aa, myc |

10 min), washed with ice-cold acetone and resuspended in SDS-PAGE loading buffer (volumes calibrated according to cell densities). Fractions were analysed by immunoblotting.

## Motility assays

For swimming motility, cultures were grown in LB broth to A600nm 1. Two microliters of culture were inoculated into soft tryptone agar (0.3% agar, 10 g/L tryptone, 5 g/L NaCl) containing ampicillin (100 µg/ml). Plates were incubated at 37 °C for between 4 and 6 hr unless otherwise stated.

## Isolation of motile strains carrying suppressor mutations

Cells of the *Salmonella flgD* null strain transformed with plasmids expressing FlgD variants (FlgDΔ2–5, FlgDΔ2-5-[19]GSGSMT[20]-$V_{15}$A or FlgD$_{short}$) were cultured at 37 °C in LB broth containing ampicillin (100 µg/ml) to mid-log phase and inoculated into soft tryptone agar (0.3% agar, 10 g/L tryptone, 5 g/L NaCl) containing ampicillin (100 µg/ml). Plates were incubated at 30 °C until motile 'spurs' appeared. Cells from the spurs were streaked to single colony and cultured to isolate the *flgD* encoding plasmid. Plasmids were transformed into the *Salmonella flgD* null strain to assess whether the plasmids were

responsible for the motile suppressor phenotypes. Plasmids were sequenced to identify the suppressor mutations.

## Quantification and statistical analysis

Experiments were performed at least three times. Immunoblots were quantified using Image Studio Lite. The unpaired two-tailed Student's $t$-test was used to determine p-values and significance was determined as $*p < 0.05$. Data are represented as mean ± standard error of the mean (SEM), unless otherwise specified and reported as biological replicates.

## Acknowledgements

This work was funded by grants from the Biotechnology and Biological Sciences Research Council (BB/M007197/1) to GMF, the Wellcome Trust (082895/Z/07/Z) to CH and GMF, a Biotechnology and Biological Sciences Research Council studentship to P.D., and a University of Cambridge John Lucas Walker studentship to OJB.

## Additional information

### Funding

| Funder | Grant reference number | Author |
|---|---|---|
| Biotechnology and Biological Sciences Research Council | BB/M007197/1 | Gillian M Fraser |
| The Wellcome Trust | | Colin Hughes |

The funders had no role in study design, data collection and interpretation, or the decision to submit the work for publication.

### Author contributions

Owain J Bryant, Conceptualization, Data curation, Formal analysis, Investigation, Methodology, Visualization, Writing – original draft, Writing – review and editing; Paraminder Dhillon, Conceptualization, Data curation, Formal analysis, Investigation, Methodology, Visualization, Writing – review and editing; Colin Hughes, Gillian M Fraser, Conceptualization, Data curation, Formal analysis, Funding acquisition, Investigation, Methodology, Project administration, Resources, Supervision, Visualization, Writing – review and editing

### Author ORCIDs

Owain J Bryant http://orcid.org/0000-0003-2683-038X
Gillian M Fraser http://orcid.org/0000-0002-4874-8734

### Decision letter and Author response

Decision letter https://doi.org/10.7554/eLife.66264.sa1
Author response https://doi.org/10.7554/eLife.66264.sa2

## Additional files

### Supplementary files

• Transparent reporting form

### Data availability

Data generated or analysed during this study are included in the manuscript and supporting files or have been submitted to Dryad (https://doi.org/10.5061/dryad.66t1g1k3x).

The following dataset was generated:

| Author(s) | Year | Dataset title | Dataset URL | Database and Identifier |
|---|---|---|---|---|
| Fraser GM | 2022 | Data from: Recognition of discrete export signals in early flagellar subunits during bacterial Type III secretion | http://dx.doi.org/10.5061/dryad.66t1g1k3x | Dryad Digital Repository, 10.5061/dryad.66t1g1k3x |

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
