## [Editor Report]

Herein, the authors show that the flagellar Type III Secretion System recognize sequentially two discrete export signals, 1. initial docking and 2. subsequent opening of the export gate, for bacterial flagella biogenesis. This important and elegantly designed study elucidates a key step in solving the long-standing question of how export substrates are recognized by the type III secretion system.

---

## [Decision Letter]

**Decision letter after peer review:**

Thank you for submitting your article "Sequential recognition of discrete export signals in flagellar subunits during bacterial Type III secretion" for consideration by *eLife*. Your article has been reviewed by 3 peer reviewers, and the evaluation has been overseen by a Reviewing Editor and Gisela Storz as the Senior Editor. The following individuals involved in review of your submission have agreed to reveal their identity: Andreas Diepold (Reviewer #1); Samuel Wagner (Reviewer #2); Chi Aizawa (Reviewer #3).

Essential revisions:

1. The general applicability of the model to all T3SS substrates would benefit from one key experiment – the removal of present small hydrophobic amino acids from the extreme N-terminus of a substrate other than FlgD, for example FliK, where only the first eight amino acids would have to be changed and the polyhook phenotype of ΔfliK might serve as an easy readout.

In the current version, the importance of the new signal (small hydrophobic amino acid) is only directly shown for one example, FlgD. The results in Figure 3 for FlgE/G only directly show an influence of the overall length of the signal upstream the GRM.

2. Is there any evidence for the formation of an α-helix in the stretch between the hydrophobic signal at the extreme N-terminus and the GRM? It should be possible to predict this based on the available export signal sequences. The fact that longer stretches do not impede export argues for an unfolded signal. Focusing on this option might allow to identify the interaction region of the new signal with more precision.

3. The discussion would benefit from a reflection of the most recent data presented by the Erhardt group on FliPQ gate opening: https://doi.org/10.1101/2020.11.25.397760

4. A bioinformatic survey of the conservation of the newly identified signal in early substrates of T3SS would be a great asset. Also, lines 396ff state that "the GRM is conserved in all four injectisome early subunits" but neither a reference nor primary data are provided.

*Reviewer #1:*

In this manuscript, Bryant and colleagues use a clever and thorough combination of genetic suppressor screens and directed mutagenesis to identify specific motifs in the N-terminal export signal for substrates of the type III secretion system (T3SS). Using the flagellar hook-cap subunit FlgD as a model export substrate, they identify a second export signal in addition to the previously identified "gate recognition motif" (GRM): the presence of a small hydrophobic amino acid at the extreme N-terminus of the substrate, located at least ~20 amino acids upstream the GRM.

The signal was identified by an extension of a previous five-residue deletion scanning screen, which allowed the identification of motile suppressor mutants from an immotile deletion of amino acids 2-5 in FlgD. The two classes of suppressors (point mutations towards small hydrophobic amino acids at the N-terminus or insertion of unspecific sequences between the first remaining small hydrophobic amino acid and the GRM) then allow to build a model for how these two signals work together, which is tested in several experiments. Sequence comparison and limited experimental data suggest that this signal is conserved throughout early substrates of the T3SS. Combinations of mutations in the new signal and the GRM suggest that the new signal acts after the GRM.

The manuscript tackles a central question in type III secretion: although an enormous number of secretion signals are known and increasingly sophisticated attempts of identifying such signals have been published, the direct molecular function of the secretion signal is still unknown. The study benefits from a clear experimental order and line of argumentation, the figures and schemes are well-composed, and the majority of results and interpretation are very convincing. In fact, based on the detailed results and the great recent progress in the characterization of the export apparatus and the T3SS membrane barrier, the authors should be able to at least make well-founded speculations on the site of interaction of their newly identified signal. Such an extended analysis, building e.g. on the two possible locations of FlhBc (the interaction site of the GRM) presented by Kuhlen et al., doi: 10.1038/s41467-020-15071-9, and the thorough characterization of the flagellar T3SS pore (a possible interaction site of the new signal) in a pre-print by Hüsing et al., doi: 10.1101/2020.11.25.397760 would make the manuscript even more valuable. At least, these new studies should be included in the interpretation and discussion.

1. The general applicability of the model to all T3SS substrates would benefit from one key experiment – the removal of present small hydrophobic amino acids from the extreme N-terminus of a substrate other than FlgD, for example FliK, where only the first eight amino acids would have to be changed and the polyhook phenotype of ΔfliK might serve as an easy readout.

In the current version, the importance of the new signal (small hydrophobic amino acid) is only directly shown for one example, FlgD. The results in Figure 3 for FlgE/G only directly show an influence of the overall length of the signal upstream the GRM.

2. Is there any evidence for the formation of an α-helix in the stretch between the hydrophobic signal at the extreme N-terminus and the GRM? It should be possible to predict this based on the available export signal sequences. The fact that longer stretches do not impede export argues for an unfolded signal. Focusing on this option might allow to identify the interaction region of the new signal with more precision.

*Reviewer #2:*

Bryant et al., have investigated the export signal that leads to secretion of rod/hook-type substrates of the type III secretion system (T3SS) of bacterial flagella using the hook-cap subunit FlgD as a model export substrate. In previous work the Fraser group showed that a hydrophobic gate recognition motif (GRM) about 40 amino acids into the protein mediates recognition of the substrate at the C-terminus of the substrate specificity switch protein FlhB.

Herein they asked whether additional information within the first 50 amino acids of the substrate is critical for secretion by analyzing a series of 5 amino acid deletion mutants. In addition to the deletion of the GRM, only the deletion of the residues 2-5 abrogated secretion, which contains a stretch of non-polar, small amino acids (IAV). Further in depth genetic analysis showed that this signal needs to be separated by 24-30 residues from the GRM for optimal functioning. Analysis of dominant negative effects of too shortly distanced (FlgDshort and others) substrate mutants indicated a sequential engagement of first the GRM with FlhBc and then the newly identified signal. Based on the currently available structural information of T3SS, the authors concluded that the N-terminal hydrophobic signal might trigger pore opening at the export gate of the system while held in place by the GRM-FlhBc interaction. They could show that a mutant compromising gate closure results in a compensation of the secretion defect of FlgDshort.

The authors contribute an important and elegantly designed piece of work to elucidate the still ill-defined secretion signals of T3SS and the mechanism of secretion itself. They discuss their findings comprehensively in the light of the state-of-the-art, which is a great asset for the reader.

While the genetic evidence is compelling, biochemical evidence is entirely lacking, for which reason the conclusions remain rather hypothetical. The statement that "sequential recognition of discrete export signals" drives gate opening is not substantiated since no temporal resolution is achieved with the experiments presented.

Furthermore, the trigger of gate opening is postulated to occur by the newly identified signal but no switch (position where signal acts) was specified.

1. The presented work could be substantially strengthened by showing an interaction between the newly identified signal and the export gate (switch). The effect of the FliPM210A mutant is nice but does not tell much about the mode-of-action of the newly identified signal.

First, a reflection of the nature of the switch would be helpful: It should possibly be of hydrophobic nature, could be within FlhA or close to the pore opening of the FliPQR-FlhB complex (likely Q or FlhBn as these are the only proteins directly accessible from the bottom).

Second, the ∆2-5 mutant could be used to biochemically show the interaction of the GRM with FlhB and of the N-terminus with the switch by in vivo photocrosslinking using the genetically encoded UV-inducible artificial amino acid pBpa.

Third, the discussion would benefit from a reflection of the most recent data presented by the Erhardt group on FliPQ gate opening: https://doi.org/10.1101/2020.11.25.397760

2. A bioinformatic survey of the conservation of the newly identified signal in early substrates of T3SS would be a great asset. Also, lines 396ff state that "the GRM is conserved in all four inectisome early subunits" but neither a reference nor primary data are provided.

*Reviewer #3:*

There are several systems of secreting proteins to the outside of the cell. This study is on a secretion system called type 3, especially on the flagellar proteins that are transported from the cytoplasm to the cell surface to make flagellar filaments. A multi-component machinery in the flagellar base located in the membrane is specialized to export flagellar proteins only. Accordingly flagellar proteins to be exported have signals to be recognized by the export machine.

Authors have been trying to identify the signals in detail using smartly designed variants. Previous results showed the one-dimensional images of N-terminal signal and now they show the 3-dimensional image of interaction between the export gate and the substrate. This work revealed a signal pattern that interacts with the machine, and thus is an important step to reveal a whole image of export events.

I have several questions before publishing.

74: Authors chose FlgD as a model protein, and they confess "it was fortuitous"(309). Please explain why it was lucky, and then FlgD should be explained a little more in the Introduction or Results, at least MW and its role.

I am wondering if this kind of experiments be done with other protein of the same category, like FlgK or FlgL.

Although these proteins belong to the late genes like FliC, some portion is expressed and secreted and their variants could stop secretion.

---

## [Author Response]

Essential revisions:1. The general applicability of the model to all T3SS substrates would benefit from one key experiment – the removal of present small hydrophobic amino acids from the extreme N-terminus of a substrate other than FlgD, for example FliK, where only the first eight amino acids would have to be changed and the polyhook phenotype of ΔfliK might serve as an easy readout.In the current version, the importance of the new signal (small hydrophobic amino acid) is only directly shown for one example, FlgD. The results in Figure 3 for FlgE/G only directly show an influence of the overall length of the signal upstream the GRM.

To address this, we deleted residues 2-8 of FliK, as suggested by the reviewers. We showed that deletion of this region,which contains small hydrophobic residues, severely reduced FliK export. These results, which further support the view that early flagellar substrates contain a hydrophobic N-terminal export signal are now shown in figure S3 and described in lines 362-365 of the manuscript.

2. Is there any evidence for the formation of an α-helix in the stretch between the hydrophobic signal at the extreme N-terminus and the GRM? It should be possible to predict this based on the available export signal sequences. The fact that longer stretches do not impede export argues for an unfolded signal. Focusing on this option might allow to identify the interaction region of the new signal with more precision.

There is evidence that the termini of the flagellar hook protein (FlgE) are disordered in solution, but this region can also form α-helical structure when assembled in the flagellum. It isn’t known whether the sequence between the hydrophobic signal and the GRM are structured or disordered when located in the environment of the export machinery but as this reviewer points out, longer stretches do not impede export which does argue that the region between both signals is unfolded. The precise location of FlhBc (the GRM docking site) in the export machinery is still unknown, however, using the AlphaFold (https://alphafold.ebi.ac.uk) structure prediction of full length FlhB and the experimentally determined structures of the export gate and basal body we have predicted the location of FlhBc in the export machinery (new Figure S10). This is based on the predicted structure of full length FlhB (which was predicted in isolation and not in the context of its other binding partners) and, therefore, this model must be interpreted with caution. The predicted model does however indicate that FlhBc is located below the plane of the inner membrane and above FlhA_C_, which indicates that when subunits are docked at FlhB_C_ the N-terminal export signal would be located within the transmembrane region of FlhA and/or at the base of the FliPQR-FlhB_N_ complex (Figure S10). Discussion of this is now included in lines 576-579 in the manuscript.

3. The discussion would benefit from a reflection of the most recent data presented by the Erhardt group on FliPQ gate opening: https://doi.org/10.1101/2020.11.25.397760

We have now included additional discussion on the most recent data presented by the Erhardt group (lines 592-608) and have referenced the paper.

4. A bioinformatic survey of the conservation of the newly identified signal in early substrates of T3SS would be a great asset. Also, lines 396ff state that "the GRM is conserved in all four injectisome early subunits" but neither a reference nor primary data are provided.

We have now included data to show the conservation of the gate recognition motif (GRM) and the newly identified signal in FlgD, FlgE and FliK early subunits (Figure S9). Discussion of this is now included in lines 542-543 in the manuscript. The N-terminal sequences of FlgD, FlgE or FliK were aligned at the GRM with N-terminal sequences of subunits from other bacterial species. We know from our data that the newly identified N-terminal signal is located at least 24 residues from the GRM and that small non-polar residues (leucine, isoleucine or valine) represent the signal. The data shows that there isn’t a consensus sequence but instead shows that the N-termini all the early subunits contain essential non-polar residues required for subunit export. We have now included an additional figure in the supplementary which shows the conservation of the GRM sequence motif in early flagellar subunits and the injectisome subunits (Figure S11).